# Selective Learning for Deep Time Series Forecasting

**Yisong Fu**[1,2], **Zezhi Shao**[1], **Chengqing Yu**[1,2], **Yujie Li**[1,2], **Zhulin An**[1,2],
**Qi Wang**[3], **Yongjun Xu**[1,2*], **Fei Wang**[1,2*]

[1]State Key Laboratory of AI Safety, Institute of Computing Technology,
Chinese Academy of Sciences   [2]University of Chinese Academy of Sciences
[3]Department of Automation, Tsinghua University

`{fuyisong24s, shaozezhi, yuchengqing22b, liyujie23s, anzhulin}@ict.ac.cn,`
`cheemswang@mail.tsinghua.edu.cn, {xyj,wangfei}@ict.ac.cn`

## Abstract

Benefiting from high capacity for capturing complex temporal patterns, deep learning (DL) has significantly advanced time series forecasting (TSF). However, deep models tend to suffer from severe overfitting due to the inherent vulnerability of time series to noise and anomalies. The prevailing DL paradigm uniformly optimizes all timesteps through the MSE loss and learns those uncertain and anomalous timesteps without difference, ultimately resulting in overfitting. To address this, we propose a novel selective learning strategy for deep TSF. Specifically, selective learning screens a subset of the whole timesteps to calculate the MSE loss in optimization, guiding the model to focus on generalizable timesteps while disregarding non-generalizable ones. Our framework introduces a dual-mask mechanism to target timesteps: (1) an uncertainty mask leveraging residual entropy to filter uncertain timesteps, and (2) an anomaly mask employing residual lower bound estimation to exclude anomalous timesteps. Extensive experiments across eight real-world datasets demonstrate that selective learning can significantly improve the predictive performance for typical state-of-the-art deep models, including 37.4% MSE reduction for Informer, 8.4% for TimesNet, and 6.5% for iTransformer.

Code: https://github.com/GestaltCogTeam/selective-learning
https://github.com/GestaltCogTeam/BasicTS

## 1 Introduction

Time series forecasting (TSF) plays a crucial role in many real-world applications, such as traffic flow prediction [26, 45, 43], weather forecasting [38, 65, 80, 12], and energy consumption planning [33, 57]. The rapid advancement of deep learning (DL) has spurred breakthroughs in TSF, with numerous deep models pushing the boundaries of predictive performance and becoming pivotal in the field [81, 64, 82, 76, 35, 29].

Despite the strong capacity for capturing complex temporal patterns, deep TSF models are prone to suffer from severe overfitting issues under certain scenarios due to the characteristics of real-world time series data [42, 58, 7]. Unlike other data modalities such as natural language and images, time series is inherently susceptible to *noise* and *anomalies* introduced by random exogenous factors [28, 56, 31]. For example, industrial sensors are easily affected by noise from mechanical vibrations and electromagnetic disturbances, and stock prices exhibit non-stationary fluctuations given the policy interventions. These interference factors are challenging to model and typically change over time, exhibiting *uncertain* and *anomalous* patterns at a specific range of timesteps. However, the current

---

[*]Corresponding authors.

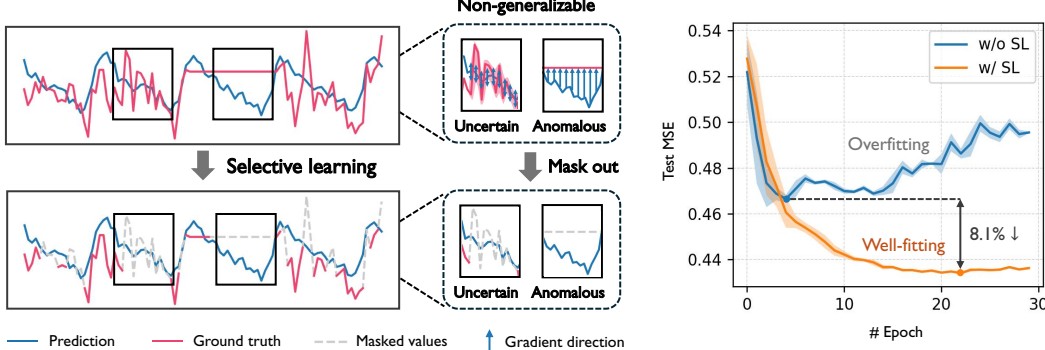

Figure 1: **Left:** When optimizing the model through MSE loss, our proposed selective learning calculates MSE only on a subset of timesteps, while masking out *uncertain* and *anomalous* ones that are *non-generalizable*. **Right:** Test MSE curves of iTransformer during training on the ETTh1 dataset (prediction length $F = 336$). The model exhibits severe overfitting, but this is effectively mitigated through selective learning, yielding an 8.1% reduction in test MSE with stable convergence.

DL paradigm treats each time step equally when training the models with regression loss functions (e.g., MSE/MAE loss). This causes an overfitting issue when forcing models to learn *uncertain* and *anomalous* timesteps unavailable to generalize, deteriorating models' performance. For example, iTransformer [29] encounters significant overfitting when trained on the ETTh1 dataset. As shown in Figure 1 (right), its test MSE during training gradually increases after the third epoch.

To address these challenges, we propose selective learning, a novel learning strategy for deep TSF. As illustrated in Figure 1 (left), the main idea of selective learning is to involve only generalizable timesteps, a subset of the time series, in optimization and discard identified *uncertain* or *anomalous* ones. In implementation, we propose a dual-mask mechanism to filter out *non-generalizable* timesteps dynamically. (1) For *uncertain* timesteps, we introduce the entropy of the prediction residual distribution as the uncertainty measure. With the sliding window sampling in time series, we can obtain multiple samples of a predicted timestep under different historical windows, thereby quantifying the entropy of the residual to serve as an indicator to filter out high-entropy ones. (2) For *anomalous* timesteps, we train an estimation model to obtain the residual lower bound of each timestep. By masking timesteps where current residuals are closest to the lower bound, it removes *non-generalizable* anomalies dynamically while keeping to-be-learned timesteps.

Extensive experiments across eight real-world datasets show that selective learning achieves consistent performance gains on six well-acknowledged deep models. It proves particularly effective for models that are susceptible to overfitting, where it achieves a **37.4%** reduction in MSE and for Informer [81], and **15.6%** in MSE for Crossformer [79]. Notably, selective learning maintains its benefits even for state-of-the-art baselines, such as TimesNet [63] (**8.4%** MSE reduction) and iTransformer [29] (**6.5%** MSE reduction). Furthermore, we conducted comparative analyses with alternative training objectives in §5.4. The consistent leading performance of selective learning further demonstrates the advantage of optimizing *generalizable* subsets over global sequence optimization.

In summary, our contribution is three-fold:

- We propose selective learning, a novel learning strategy for deep TSF to identify and expel *non-generalizable* timesteps in optimization. It is the first trial to address the overfitting issue from the timestep granularity in the field.

- Technically, we devise a tractable strategy by introducing a dual-mask mechanism to filter out *uncertain* and *anomalous* timesteps dynamically during training.

- Our method is agnostic to deep learning backbones and examined across several real-world datasets. The results demonstrate its effectiveness and show consistent performance improvement over all baselines.

## 2 Related Work

### 2.1 Deep Models for Time Series Forecasting

In recent years, numerous deep models have been proposed to capture complex dependencies in TSF. Transformer-based models have gained significant attention for their ability to capture long-term temporal dependencies through attention mechanisms [81, 64, 82, 74, 79, 35, 29]. For example, Informer [81] introduces a ProbSparse attention to reduce the quadratic complexity. PatchTST [35] splits time series into patches and employs a channel-independent strategy. iTransformer [29] embeds each series independently to the variate token and applies self-attention to capture multivariate correlations. In contrast to Transformers, CNN-based models [63, 32, 25] exhibit strong proficiency in extracting local patterns. Typically, TimesNet [63] transforms time series into 2D tensors and employs CNN to capture inter- and intra-period dependencies. Additionally, MLP-based models offer efficient alternatives with lightweight architectures [44, 13, 55, 70, 76]. DLinear [76] leverages a simple linear layer with decomposition, and TimeMixer [55] captures multi-scale information through MLP layers. Our proposed selective learning can be easily applied to these deep models.

### 2.2 Training Strategies for Time Series Forecasting

The prevailing DL paradigm computes the regression loss (e.g., MSE/MAE) uniformly across all timesteps, and some works have explored alternative training strategies. For example, iTransformer [29] proposes a training strategy that randomly selects subsets of variables for large-scale multivariate time series. Merlin [75] employs a knowledge distillation [15] strategy to enhance the model's robustness against data missing. These approaches are optimized for specific scenarios or tightly coupled with model architectures, limiting their broader applicability. The most relevant work is MTGNN [67], which applies the idea of curriculum learning [4] to TSF by progressively increasing the prediction length during training. However, it overlooks that difficulty is not solely determined by prediction length but is also influenced by intrinsic data characteristics. Selective learning addresses this issue by masking *non-generalizable* timesteps while maintaining broad applicability.

Another line of work proposes alternative training objectives to replace the regression loss. For example, Soft-DTW [8], DILATE [23], and TILDE-Q [24] align the shape between predictions and target sequences under temporal distortions. FreDF [51] combines the MSE loss with a frequency loss, mitigating the label correlation. PS loss [21] enhances the alignment by incorporating patch-wise distribution information. These works focus on matching the shape or distribution in temporal or frequency domain between sequences, but none have recognized that global alignment over the whole sequence is not optimal, as certain timesteps in the target sequence are inherently *non-generalizable*.

## 3 Preliminaries

**Notations** For a multivariate time series with $N$ variables, let $X_t \in \mathbb{R}^N$ represent the $t$-th timestep. Given a historical time series $\mathbf{X}_{t-L:t} = \{X_{t-L}, X_{t-L+1}, \cdots, X_{t-1}\} \in \mathbb{R}^{L \times N}$, where $L$ is the look-back window size, the TSF task is to predict future values $\hat{\mathbf{X}}_{t:t+F} = \{X_t, X_{t+1}, \cdots, X_{t+F-1}\} \in \mathbb{R}^{F \times N}$ with forecasting window size $F$. Considering a historical time series $\mathbf{X}_{0:T} \in \mathbb{R}^{T \times N}$ for training, the training dataset $\mathcal{D}_{train} = \{(\mathbf{X}_{t-L:t}, \mathbf{X}_{t:t+F})\}_{t=L}^{T-F}$ is constructed by a sliding window approach with stride 1.

**Problem Statement** The current DL paradigm is to find the best mapping from the samples in $\mathcal{D}_{train}$, i.e., $\hat{\mathbf{X}}_{t:t+F} = f(\mathbf{X}_{t-L:t}; \boldsymbol{\theta})$, where $f(\cdot; \boldsymbol{\theta}) : \mathbb{R}^{L \times N} \to \mathbb{R}^{F \times N}$ is a deep neural network parameterized by $\theta$. Mean squared error (MSE) measures the discrepancy between the prediction $\hat{\mathbf{X}}_{t:t+F}$ and the ground truth $\mathbf{X}_{t:t+F}$ and is one of the commonly used loss functions to optimize $\boldsymbol{\theta}$:

$$\mathcal{L}_{MSE}(\boldsymbol{\theta}) = \frac{1}{N \cdot F} \sum_{i=0}^{F-1} ||X_{t+i} - f(\mathbf{X}_{t-L:t}; \boldsymbol{\theta})_i||^2, \tag{1}$$

$$\boldsymbol{\theta}_{\tau+1} = \boldsymbol{\theta}_\tau - \eta \nabla_{\boldsymbol{\theta}} \mathcal{L}_{MSE}, \tag{2}$$

where $\eta$ is the learning rate. We use $\tau$ to denote the number of iterations during training, distinguishing it from the timestep index $t$.

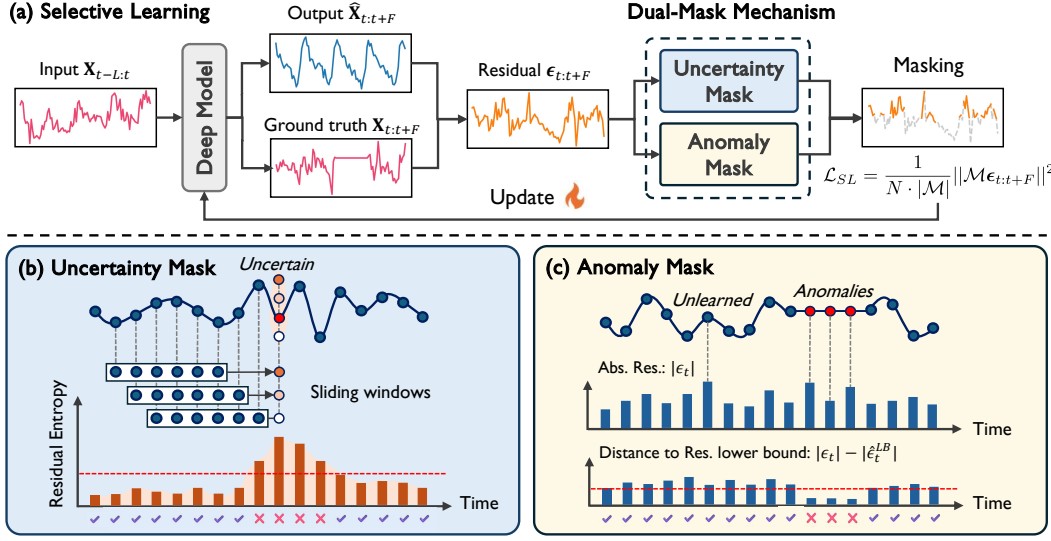

Figure 2: (a) Overall framework of selective learning. (b) Uncertainty mask. (c) Anomaly mask.

# 4 Selective Learning

## 4.1 Overview of Selective Learning

We propose selective learning, a model-agnostic learning strategy for deep TSF to address the overfitting issue. The main idea of selective learning is to calculate the MSE loss only on a subset of timesteps. This enables models to focus selectively on *generalizable* timesteps while disregarding *non-generalizable* ones. Figure 2 (a) illustrates the overall framework of selective learning. The implementation details and workflow of selective learning are provided in Appendix C.3.

As illustrated in Figure 1 (left), we identify two critical categories of timesteps that degrade model generalizability. (1) *Uncertain* timesteps. Primarily originating from inherent noise in time series (e.g., signal disturbances [23]), they are characterized by high predictive uncertainty. Consequently, the gradients at these timesteps update towards random directions, resulting in undesirable fitting to the noise. (2) *Anomalous* timesteps. They are mainly caused by exogenous exceptional events (e.g., sensor malfunctions [73, 72]). The model's predictions, though possibly confident, exhibit significant errors. This forces the model to learn instance-specific features through biased gradient updates, ultimately harming generalization.

To this end, we propose a dual-mask mechanism to dynamically filter out these non-generalizable timesteps, as shown in Figure 2. Formally, we define selective learning as follows: Given a deep TSF model $f(\cdot, \boldsymbol{\theta})$ at $\tau$-th iteration, we find a mask $\mathcal{M}^{(\tau)} \in \{0, 1\}^F$ that constrains optimization only over a subset of timesteps:

$$\mathcal{L}_{SL}(\boldsymbol{\theta}) = \frac{1}{N \cdot |\mathcal{M}^{(\tau)}|} \sum_{i=0}^{F-1} ||\mathcal{M}^{(\tau)}(X_{t+i} - f(\mathbf{X}_{t-L:t}; \boldsymbol{\theta})_i)||^2, \tag{3}$$

$$\boldsymbol{\theta}_{\tau+1} = \boldsymbol{\theta}_\tau - \eta \nabla_{\boldsymbol{\theta}} \mathcal{L}_{SL}, \tag{4}$$

where $\mathcal{M}^{(\tau)} = \mathcal{M}_u^{(\tau)} \vee \mathcal{M}_a^{(\tau)}$, and $\vee$ is the element-wise OR operator. $\mathcal{M}_u^{(\tau)}, \mathcal{M}_a^{(\tau)} \in \{0, 1\}^F$ are the uncertainty mask and anomaly mask, respectively. We will describe them in detail in the following sections. Since uncertain and anomalous patterns do not necessarily appear synchronously across all variables, we adopt the channel-independent strategy [35], generating masks for each variable independently. For theoretical tractability, we will focus on the univariate case in subsequent analysis, with natural extensibility to multivariate scenarios due to channel independence.

## 4.2 Uncertainty Mask

We propose an entropy-based uncertainty masking approach for those timesteps that exhibit high predictive uncertainty. Let $\epsilon_t = X_t - \hat{X}_t$ denote the residual of $t$-th timesteps. The differential entropy of $\epsilon_t$ is

$$H(\epsilon_t) = \int p(\epsilon_t) \ln p(\epsilon_t) \mathrm{d}t. \tag{5}$$

Since training samples are constructed via sliding windows over $\mathbf{X}_{1:T}$, each timestep will be predicted $n_t$ times in one epoch, where $n_t = \min\{t - L + 1, F\}$. Therefore, we can estimate the residual distribution of $\epsilon_t$ using the most recent $n_t$ predictions. Let $l(\boldsymbol{\psi}|\epsilon_t)$ be the likelihood model of the residual, and we have

$$\hat{\boldsymbol{\psi}} = \arg\max_{\boldsymbol{\psi}} l(\boldsymbol{\psi}|\epsilon_t^{(1)}, \epsilon_t^{(2)}, \cdots, \epsilon_t^{(n_t)}), \tag{6}$$

$$\hat{H}(\epsilon_t) = \int l(\hat{\boldsymbol{\psi}}|\epsilon_t) \ln l(\hat{\boldsymbol{\psi}}|\epsilon_t) \mathrm{d}t, \tag{7}$$

where $\epsilon_t^{(i)}$ is the $i$-th most recent prediction residual.

In practice, we assume that the residual $\epsilon_t \sim \mathcal{N}(\mu_t, \sigma_t^2)$, therefore we have

$$\hat{H}(\epsilon_t) = \frac{1}{2} \ln(2\pi e \hat{\sigma}_t^2), \tag{8}$$

$$\hat{\sigma}_t^2 = \frac{1}{n_t} \sum_{i=1}^{n_t} (\epsilon_t^{(i)} - \bar{\epsilon}_t)^2 \tag{9}$$

Since these residuals $\epsilon_t^{(i)}$ are computed at different training time $\tau$, they originate from distinct $f(\cdot, \boldsymbol{\theta}_\tau)$. Under Assumptions 2-4 (Appendix A.1), Theorem 1 provides an upper bound for the error introduced by different $\boldsymbol{\theta}$.

**Theorem 1** (Upper Bound for Variance Estimation Error ). *The error bound between variance estimation under distinct parameters $\hat{\sigma}_t^2$ and that under identical parameters $\hat{\sigma}_t^2(\boldsymbol{\theta}_\tau)$ satisfies:*

$$|\hat{\sigma}_t^2 - \hat{\sigma}_t^2(\boldsymbol{\theta}_\tau)| \leq 4L_f R\eta G(2K-1), \tag{10}$$

*where $K$ is the number of iterations per epoch, and $L_f, R, G$ are constants.*

The proof is provided in Appendix A. According to Theorem 1, we can always control the estimation error by choosing a sufficiently small learning rate $\eta$ and a large batch size.

We employ a hard thresholding $\gamma_u$ on the top-$r_u\%$ residual entropy and obtain the uncertainty mask $\mathcal{M}_u^{(\tau)} \in \{0,1\}^F$ satisfying

$$(\mathcal{M}_u^{(\tau)})_t = \begin{cases} 0, & \hat{H}(\epsilon_t) > \gamma_u, \\ 1, & \text{otherwise.} \end{cases} \tag{11}$$

## 4.3 Anomaly Mask

Predictions for anomalous timesteps typically exhibit significantly larger residuals due to deviations in ground truth values. The most intuitive solution is to mask timesteps with high $|\epsilon_t|$ to filter anomalies. However, this naive approach suffers from a critical limitation: it indiscriminately excludes both genuine anomalies and currently unlearned yet (but potentially generalizable) patterns, particularly during the early stage of training when the learning process remains incomplete.

To overcome this limitation, we draw inspiration from practices in other fields [34, 48, 27, 50], and define $S(X_t)$ as the deviation between the residual and its theoretical lower bound $\epsilon_t^{LB}$:

$$S(X_t) = |X_t - f(\mathbf{X}; \boldsymbol{\theta})_t)| - \epsilon_t^{LB}. \tag{12}$$

This formulation enables a key separation: Anomalous timesteps exhibit elevated residual lower bounds, resulting in comparatively small $S(X)$ values. In contrast, unlearned timesteps demonstrate larger $S(X)$ values due to significant gaps between current residuals and their theoretical minima.

In practice, we train a lightweight model $g(\cdot; \phi)$ on $\mathcal{D}_{train}$ to estimate the residual lower bound, thereby estimating $S(X_t)$ by:

$$\hat{S}(X_t) = \underbrace{|X_t - f(\mathbf{X}; \boldsymbol{\theta})_t|}_{\text{residual } \epsilon_t} - \underbrace{|X_t - g(\mathbf{X}; \boldsymbol{\phi})_t|}_{\text{estimated LB } \hat{\epsilon}_t^{LB}}. \tag{13}$$

The details of the estimation model are discussed in Appendix C.3. Analogous to the uncertainty mask, we employ a hard thresholding $\gamma_a$ to filter out the top-$r_u\%$ of timesteps with the smallest $\hat{S}(X_t)$ and obtain the anomaly mask $\mathcal{M}_a^{(\tau)} \in \{0, 1\}^F$:

$$(\mathcal{M}_a^{(\tau)})_t = \begin{cases} 0, & \hat{S}(X_t) < \gamma_a, \\ 1, & \text{otherwise.} \end{cases} \tag{14}$$

Notably, instead of using the estimated residual lower bound as a static masking criterion, $S(X_t)$ can dynamically adjust the masking based on the current predictions. This approach offers two key advantages: (1) Static masking significantly alters the distribution of $\mathcal{D}_{train}$, thereby introducing bias, whereas dynamic masking adapts the mask during training to mitigate this in expectation. (2) For rare but critical extreme events (e.g., extreme weather) [10, 78] that are less generalizable, dynamic masking first learns the most generalizable timesteps and gradually attempts to learn timesteps previously considered anomalies. See Appendix E.2 for detailed discussions.

## 5 Experiments

### 5.1 Experimental Setup

**Datasets**   We thoroughly evaluate the effectiveness of the proposed selective learning on 8 real-world datasets, including Electricity, Exchange, Weather, ILI, and 4 ETT datasets (ETTh1, ETTh2, ETTm1, ETTm2), which have been extensively used for benchmarking [42, 64, 37]. A detailed description of the datasets is provided in Appendix C.1.

**Baselines**   Selective learning is a model-agnostic training strategy, and it is compatible with any deep TSF models. We carefully select six well-acknowledged deep models as the baselines, including Transformer-based models (Informer [81], Crossformer [79], PatchTST [35], iTransformer [29]), CNN-based models (TimesNet [63]), and MLP-based models (TimeMixer [55]). See Appendix C.2 for the introduction to the baselines. We select DLinear [76] as the estimation model for all baselines, and we further discuss the effects of different estimation models in §5.5.

**Experimental Settings**   All baselines follow the same experimental setup with prediction lengths $F \in \{24, 36, 48, 60\}$ for ILI and $F \in \{96, 192, 336, 720\}$ for others [63]. We search for the lookback window $L$ and report the best results. For fair evaluation, when training baselines with selective learning to enhance their performance, we follow their original hyperparameter settings and only tune the masking ratios $r_a$ and $r_n$. We utilize Adam [20] for the model optimization. We evaluate the performance of all baselines using two commonly used metrics, MSE and MAE. All experiments are implemented with PyTorch and conducted on 8 NVIDIA GeForce RTX 4090 24GB GPUs.

### 5.2 Main Results

Table 1 shows the forecasting results with and without selective learning. The results are averaged over three runs. The lower MSE/MAE indicates a more accurate prediction. Our comprehensive evaluations demonstrate that selective learning consistently enhances model performance in all 192 cases (see full results in Appendix G.1). Selective learning proves particularly impactful for early-generation architectures that are susceptible to overfitting, where it achieves an average reduction of **37.4%** in MSE and **25.4%** in MAE for Informer [81] (**66.8%** MSE and **42.6%** MAE reduction in the ETTm2 dataset), and **15.6%** in MSE and **10.5%** in MAE for Crossformer [79]. Notably, it maintains its benefits even for state-of-the-art baselines, such as iTransformer [29] (**6.5%** MSE and **4.4%** MAE reduction) and TimeMixer [55] (**4.3%** MSE and **3.3%** MAE reduction), where the improvements persist in models already equipped with RevIN [19]. This confirms that selective learning provides additional performance gains over existing distribution shift mitigation techniques.

Table 1: Comparison of forecasting results without/with selective learning (SL). We use prediction lengths $F \in \{24, 36, 48, 60\}$ for ILI and $F \in \{96, 192, 336, 720\}$ for other datasets. Results are averaged from all prediction lengths. Better results are in **bold**, and $\Delta$ denotes the improvements caused by selective learning. Full results of ETTh2 and ETTm1 are provided in Appendix G.1.

| | ETTh1 | | ETTm2 | | Electricity | | Exchange | | Weather | | ILI | |
|---|---|---|---|---|---|---|---|---|---|---|---|---|
| Method | MSE | MAE | MSE | MAE | MSE | MAE | MSE | MAE | MSE | MAE | MSE | MAE |
| Informer | 1.289 | 0.917 | 1.485 | 0.919 | 0.342 | 0.420 | 1.520 | 0.985 | 0.337 | 0.374 | 4.953 | 1.544 |
| +SL | **0.538** | **0.534** | **0.494** | **0.527** | **0.292** | **0.386** | **0.814** | **0.712** | **0.273** | **0.299** | **3.997** | **1.360** |
| $\Delta$ | -58.3% | -41.8% | -66.8% | -42.6% | -14.4% | -8.21% | -46.4% | -27.8% | -19.1% | -20.1% | -19.3% | -11.9% |
| Crossformer | 0.455 | 0.465 | 0.588 | 0.528 | 0.182 | 0.277 | 0.755 | 0.649 | 0.226 | 0.284 | 3.982 | 1.342 |
| +SL | **0.431** | **0.441** | **0.370** | **0.408** | **0.168** | **0.264** | **0.527** | **0.525** | **0.213** | **0.265** | **3.681** | **1.285** |
| $\Delta$ | -5.33% | -5.22% | -37.1% | -22.7% | -7.71% | -4.87% | -30.1% | -19.2% | -5.97% | -6.61% | -7.56% | -4.26% |
| PatchTST | 0.427 | 0.433 | 0.271 | 0.329 | 0.167 | 0.262 | 0.342 | 0.396 | 0.228 | 0.262 | 2.076 | 0.921 |
| +SL | **0.410** | **0.417** | **0.252** | **0.312** | **0.165** | **0.258** | **0.337** | **0.384** | **0.225** | **0.250** | **1.905** | **0.895** |
| $\Delta$ | -4.10% | -3.70% | -6.83% | -5.32% | -1.05% | -1.53% | -1.46% | -3.10% | -1.32% | -4.77% | -8.26% | -2.74% |
| TimesNet | 0.499 | 0.486 | 0.289 | 0.343 | 0.198 | 0.301 | 0.382 | 0.429 | 0.248 | 0.284 | 2.493 | 1.028 |
| +SL | **0.429** | **0.439** | **0.258** | **0.316** | **0.191** | **0.294** | **0.363** | **0.413** | **0.239** | **0.271** | **2.154** | **0.931** |
| $\Delta$ | -14.0% | -9.67% | -10.7% | -7.74% | -3.29% | -2.33% | -4.97% | -3.73% | -3.54% | -4.58% | -13.6% | -9.36% |
| iTransformer | 0.458 | 0.457 | 0.273 | 0.332 | 0.164 | 0.257 | 0.364 | 0.413 | 0.235 | 0.269 | 1.909 | 0.914 |
| +SL | **0.415** | **0.425** | **0.256** | **0.313** | **0.157** | **0.249** | **0.343** | **0.399** | **0.229** | **0.257** | **1.710** | **0.857** |
| $\Delta$ | -9.29% | -6.90% | -6.40% | -5.51% | -4.27% | -2.92% | -5.78% | -3.45% | -2.87% | -4.28% | -10.4% | -6.24% |
| TimeMixer | 0.443 | 0.445 | 0.265 | 0.323 | 0.163 | 0.259 | 0.348 | 0.400 | 0.230 | 0.276 | 2.163 | 0.932 |
| +SL | **0.411** | **0.421** | **0.251** | **0.309** | **0.160** | **0.254** | **0.335** | **0.394** | **0.226** | **0.268** | **2.026** | **0.895** |
| $\Delta$ | -7.11% | -5.40% | -5.01% | -4.26% | -2.15% | -2.12% | -3.60% | -1.38% | -1.74% | -2.81% | -6.37% | -4.06% |

## 5.3 Zero-shot Forecasting

We conducted zero-shot forecasting experiments to evaluate the generalization benefits of selective learning across different datasets. Following prior works [83, 17, 5], we trained the models on dataset $\mathcal{D}_A$ and assessed on unseen dataset $\mathcal{D}_B$ without further training. As shown in Table 2, selective learning consistently enhance the performance of the baselines across diverse datasets in zero-shot forecasting, demonstrating its generalization advantage. Notably, in challenging generalization scenarios (ETTh2→ETTh1 and ETTm2→ETTm1), selective learning achieves significant improvements, with MSE reduced by **22.6%** and MAE by **14.5%** on average. Furthermore, in cases like ETTh1→ETTh2 and ETTm1→ETTm2, the results outperforms training from scratch on the target dataset, underscoring the benefits of selective learning.

Table 2: Zero-shot forecasting results on ETT datasets without/with selective learning. $\mathcal{D}_A \to \mathcal{D}_B$ denotes that the model was trained on $\mathcal{D}_A$ and tested on $\mathcal{D}_A$. The results are averaged from all prediction lengths. Better results are in **bold**, and red indicates a better result than training from scratch on $\mathcal{D}_B$ without selective learning.

| Method | TimesNet | | | | iTransformer | | | | TimeMixer | | | |
|---|---|---|---|---|---|---|---|---|---|---|---|---|
| | w/o | | **+SL** | | w/o | | **+SL** | | w/o | | **+SL** | |
| Metric | MSE | MAE | MSE | MAE | MSE | MAE | MSE | MAE | MSE | MAE | MSE | MAE |
| ETTh1→ETTh2 | 0.469 | 0.465 | **0.419** | **0.439** | 0.421 | 0.434 | **0.394** | **0.420** | 0.424 | 0.438 | **0.389** | **0.416** |
| ETTh1→ETTm2 | 0.359 | 0.397 | **0.351** | **0.389** | 0.322 | 0.371 | **0.310** | **0.359** | 0.313 | 0.360 | **0.300** | **0.352** |
| ETTh2→ETTh1 | 0.828 | 0.642 | **0.595** | **0.517** | 0.622 | 0.555 | **0.501** | **0.487** | 0.679 | 0.568 | **0.560** | **0.510** |
| ETTm1→ETTh2 | 0.483 | 0.485 | **0.465** | **0.469** | 0.447 | 0.456 | **0.428** | **0.444** | 0.441 | 0.453 | **0.434** | **0.445** |
| ETTm1→ETTm2 | 0.321 | 0.363 | **0.288** | **0.339** | 0.276 | 0.334 | **0.271** | **0.326** | 0.275 | 0.326 | **0.267** | **0.319** |
| ETTm2→ETTm1 | 0.727 | 0.578 | **0.459** | **0.448** | 0.554 | 0.498 | **0.443** | **0.439** | 0.451 | 0.466 | **0.428** | **0.425** |

## 5.4 Comparison with Other Training Objectives

Our proposed selective learning exhibits strong compatibility. It is completely model-agnostic and can be applied to any deep learning architecture with various normalization methods [19, 11, 14]. Moreover, it maintains compatibility with learning strategies such as curriculum learning for TSF

Table 3: Comparison between selective learning (SL) and other training objectives with iTransformer as backbone. The results are averaged from all prediction lengths. The best results are in **bold**, and the second-best are underlined. Full results are provided in Appendix G.2.

| Training objective | SL | | PS | | FreDF | | TILDE-Q | | MSE | |
|---|---|---|---|---|---|---|---|---|---|---|
| Metric | MSE | MAE | MSE | MAE | MSE | MAE | MSE | MAE | MSE | MAE |
| ETTh1 | **0.415** | **0.425** | 0.427 | 0.440 | 0.450 | 0.455 | 0.432 | 0.439 | 0.458 | 0.457 |
| ETTm2 | **0.257** | **0.315** | 0.264 | 0.320 | 0.262 | 0.319 | 0.263 | 0.319 | 0.273 | 0.332 |
| Exchange | **0.343** | **0.399** | 0.366 | 0.409 | 0.376 | 0.413 | 0.369 | 0.414 | 0.364 | 0.413 |
| Weather | **0.229** | **0.257** | 0.233 | 0.265 | 0.239 | 0.274 | 0.232 | 0.262 | 0.235 | 0.269 |

[67, 42]. However, selective learning operates on point-wise training objectives. In this section, we compare our selective learning with alternative non-point-wise training objectives, including shape-based (TILDE-Q [24]), frequency-based (FreDF [51]), and distribution-based (PS loss [21]) objectives. As shown in Table 3, selective learning consistently achieves superior performance. These results demonstrate that training the model by global alignment over whole sequences, whether in shape or distribution, in the temporal or frequency domain, proves suboptimal, validating the effectiveness of selective learning.

## 5.5 Ablation Study and Hyperparameter Analysis

**Ablation Study**   To study the effectiveness of the components of selective learning, we conduct an ablation study covering: (1) removing either mask from the dual-mask mechanism, and (2) replacing the dual-mask mechanism with random masking with the same masking ratio. The results in Table 4 demonstrate that the model with full selective learning consistently achieves the best performance. Removing either mask leads to significant performance degradation across all four datasets, indicating that both masks contribute essential and distinct functionalities in filtering out non-generalizable patterns. Additionally, replacing the dual-mask mechanism with random masking reduces model performance to levels comparable to or worse than the unmasked counterparts. This suggests that randomly attending to a subset of timesteps fails to enhance the model's performance and generalizability. The effectiveness of selective learning fundamentally stems from our dual-mask mechanism, which directs model attention to *generalizable* timesteps while filtering out *non-generalizable* ones.

Table 4: Ablation results for selective learning with iTransformer as backbone. The results are averaged from all predicted lengths. Full ablation results are provided in Appendix G.3.

| Dataset | | ETTh1 | | ETTm2 | | Electricity | | Weather | |
|---|---|---|---|---|---|---|---|---|---|
| Metric | | MSE | MAE | MSE | MAE | MSE | MAE | MSE | MAE |
| **Selective Learning** | | **0.415** | **0.425** | **0.257** | **0.315** | **0.157** | **0.249** | **0.229** | **0.257** |
| w/o | Uncertainty mask | 0.436 | 0.443 | 0.265 | 0.322 | 0.162 | 0.256 | 0.232 | 0.266 |
| | Anomaly mask | 0.431 | 0.438 | 0.266 | 0.323 | 0.159 | 0.252 | 0.234 | 0.267 |
| Replace | Random mask | 0.457 | 0.460 | 0.274 | 0.332 | 0.165 | 0.261 | 0.237 | 0.269 |

**Effects of Masking Ratio**   After validating the effectiveness of the dual-mask mechanism through ablation studies, we further investigate the effects of the masking ratios. When investigating a particular mask, we vary its masking ratio while fixing the other masking ratio at 0. The results are shown in Figure 3. We can observe that larger masking ratios demonstrate superior performance on highly non-stationary datasets (ETTh1 and Exchange). This indicates severe overfitting in such datasets, where models benefit from focusing selectively on the most generalizable patterns. In contrast, datasets exhibiting periodic patterns (Weather) show improved performance with smaller masking ratios. Besides, it can be observed that on the Exchange dataset, the 90% anomaly masking ratio yields peak performance. This occurs because market-induced non-generalizable anomalies in this dataset exert significantly greater influence than noise. However, in most scenarios, the best

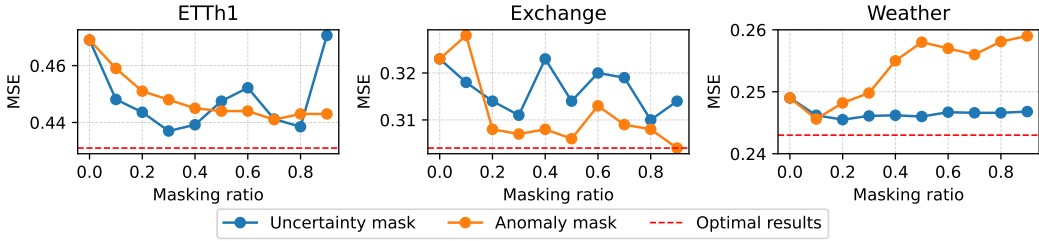

Figure 3: Forecasting results under different masking ratios. The prediction length is 336.

results are achieved by a combination of two masking strategies, the ratios of which constitute critical hyperparameters of selective learning. Detailed guidelines for selecting optimal ratios are provided in the appendix C.4.

**Effects of Estimation Model**    In the experiments above, we employed DLinear as the lightweight estimation model. In this section, we investigate the effects of different estimation models. We fix the iTransformer as the backbone model and additionally compare three estimation models: MLP, TimeMixer, and iTransformer. The results are shown in Figure 4. We can observe that: For highly non-stationary datasets (ETTh1 and Exchange), simpler models demonstrate superior performance; Conversely, datasets exhibiting periodic patterns (Weather) benefit from more complex estimation models. Despite this, the choice of the estimation model has a limited overall impact on performance, underscoring the robustness of the selective learning.

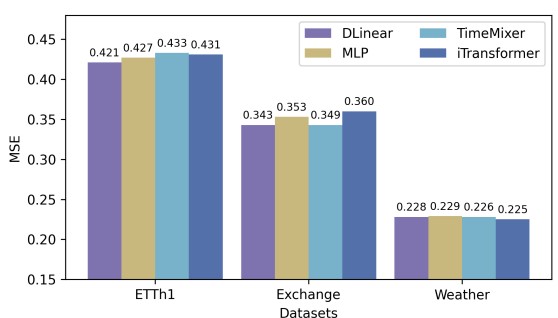

Figure 4: Forecasting performance with iTransformer as backbone and various estimation models. The results are averaged from all prediction lengths.

## 5.6   Learning Curve Analysis

In Figure 1, we initially illustrated iTransformer's training dynamics, highlighting selective learning's capacity to mitigate overfitting. To further validate this, Figure 5 presents additional learning curves on the ETTh1 dataset. While all three models exhibit varying degrees of overfitting, their counterparts trained with selective learning achieve stable convergence and superior performance. This demonstrates the efficacy of selective learning in mitigating overfitting and enhancing generalizability.

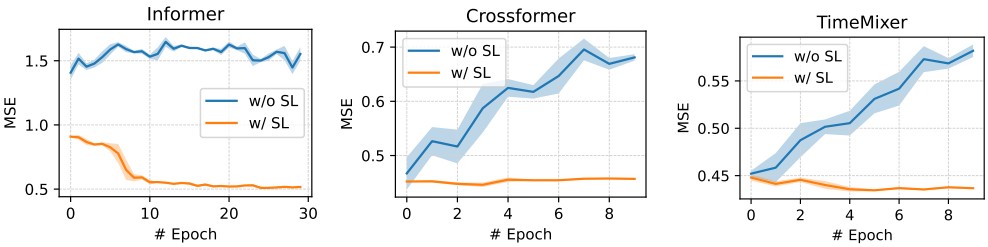

Figure 5: Test MSE curve on the ETTh1 dataset. The prediction length is 336.

# 6 Conclusion

In this work, we introduced selective learning, a novel strategy to mitigate overfitting in deep TSF by selectively computing regression loss on generalizable timesteps. Our dual-mask mechanism, comprising an uncertainty mask (based on residual entropy) and an anomaly mask (leveraging residual lower-bound estimation), dynamically filters non-generalizable timesteps, allowing models to focus on robust patterns. Extensive experiments across eight real-world datasets validate that selective learning improves predictive accuracy and model generalizability. The scope of this work is currently constrained to in-domain time series forecasting. Future work can investigate the generalization to diverse time series analysis tasks (e.g., classification, imputation) and explore pretraining strategies for time series foundation models. See Appendix F for limitations discussion and future directions.

## Acknowledgement

This work is supported by the NSFC underGrant Nos.62372430 and 62502505, the Youth Innovation Promotion Association CAS No.2023112, the Postdoctoral Fellowship Program of CPSF under Grant Number GZC20251078, the China Postdoctoral Science Foundation No.2025M77154 and HUA Innovation fundings. We sincerely thank all the anonymous reviewers who gerenously contributed their time and efforts.

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

# A Proofs

## A.1 Assumptions

**Assumption 2** (Lipschitz Continuity). *We assume that $f$ reserves the Lipschitz continuity w.r.t. $\boldsymbol{\theta}$, i.e., $\forall \boldsymbol{\theta}_1, \boldsymbol{\theta}_2 \in \boldsymbol{\Theta}$ satisfying*

$$||f(\mathbf{X}; \boldsymbol{\theta}_1) - f(\mathbf{X}; \boldsymbol{\theta}_2)|| \leq L_f ||\boldsymbol{\theta}_1 - \boldsymbol{\theta}_2||, \tag{15}$$

*where $L_f$ is the Lipschitz constant.*

**Justification** It can be ensured by the Lipschitz-continuous activations of the neural network and the continuously differentiable MSE loss.

**Assumption 3** (Bounded Prediction Residual). *We assume that there exists a constant $R > 0$ such that*

$$|\epsilon_t| \leq R, \quad \forall t \in \{1, \ldots, T\}. \tag{16}$$

**Justification** Empirically, residuals often follow a light-tailed distribution, where extreme deviations are rare. Therefore, there exists a finite high-probability bound.

**Assumption 4** (Bounded Gradient). *We assume that there exists a constant $G > 0$ such that*

$$||\nabla_{\boldsymbol{\theta}_\tau} \mathcal{L}|| < G, \quad \forall \tau \in \mathbb{Z}^+. \tag{17}$$

**Justification** The boundedness of gradients is ensured by Lipschitz-continuous activations and weight constraints of the neural network, preventing explosive updates and ensuring stable optimization. Gradient clipping can further enforce it.

## A.2 Proof of Theorem 1

*Proof.* Let $\tau_i$ denote the training time corresponding to the i-th most recent prediction residual $\epsilon_t^{(i)}$, such that $\epsilon_t(\boldsymbol{\theta}_{\tau_i}) \equiv \epsilon_t^{(i)}$.

$$
\begin{aligned}
|\hat{\sigma}_t^2 - \hat{\sigma}_t^2(\boldsymbol{\theta}_\tau)| &\leq \frac{1}{n_t} \sum_{i=1}^{n_t} |\epsilon_t^2(\boldsymbol{\theta}_{\tau_i}) - \epsilon_t^2(\boldsymbol{\theta}_\tau) - n_t(\bar{\epsilon}_t^2 - \bar{\epsilon}_t^2(\boldsymbol{\theta}_\tau))| \\
&\leq \frac{1}{n_t} \sum_{i=1}^{n_t} |\epsilon_t^2(\boldsymbol{\theta}_{\tau_i}) - \epsilon_t^2(\boldsymbol{\theta}_\tau)| + |\bar{\epsilon}_t^2 - \bar{\epsilon}_t^2(\boldsymbol{\theta}_\tau)| \\
&\leq 2 \max_i |\epsilon_t^2(\boldsymbol{\theta}_{\tau_i}) - \epsilon_t^2(\boldsymbol{\theta}_\tau)| \\
&\leq 2 \max_i |\epsilon_t(\boldsymbol{\theta}_{\tau_i}) - \epsilon_t(\boldsymbol{\theta}_\tau)| \cdot |\epsilon_t(\boldsymbol{\theta}_{\tau_i}) + \epsilon_t(\boldsymbol{\theta}_\tau)| \\
&= 2 \max_i |f(\mathbf{X}, \boldsymbol{\theta}_{\tau_i})_t - f(\mathbf{X}, \boldsymbol{\theta}_\tau)_t| \cdot |\epsilon_t(\boldsymbol{\theta}_{\tau_i}) + \epsilon_t(\boldsymbol{\theta}_\tau)| \\
&\leq 2 \max_i L_f ||\boldsymbol{\theta}_{\tau_i} - \boldsymbol{\theta}_\tau|| \cdot 2R \quad \text{(Assumption 2 and 3)} \\
&\leq 4 L_f R \eta G \max_i |\tau - \tau_i|. \quad \text{(Assumption 4)}
\end{aligned}
\tag{18}
$$

Given that adjacent epochs are separated by no more than $2K - 1$ iterations, we obtain:

$$|\hat{\sigma}_t^2 - \hat{\sigma}_t^2(\boldsymbol{\theta}_\tau)| \leq 4 L_f R \eta G (2K - 1). \tag{19}$$

$\square$

# B    Related Work

## B.1    Data Selection Strategies in Deep Learning

Data selection strategies aim to improve the efficiency, generalizability, and robustness of deep learning models by carefully selecting subsets of data for training. Instead of using the entire dataset indiscriminately, these methods prioritize samples that contribute more significantly to model learning dynamics or downstream performance. Some studies such as curriculum learning [4] and hard sample mining [47], select or filter samples based on their *difficulties*. In LM pretraining, samples or tokens are typically selected for high *quality* [68, 61, 27, 3], great *importance* [69], or strong *diversity* [77] to enhance both training efficiency and generalization. Data filtering pipelines often rely on heuristic metrics, such as perplexity [60, 49] and toxicity, or learned metrics [34, 27, 48], to remove duplicated, or low-quality data. Unlike the above studies, our approach is closely tied to the characteristics of time series data, focusing on filtering noise or anomalies. By identifying and removing such irregular patterns, the model can learn from representative and generalizable timesteps and achieve more stable and reliable predictions.

# C    Implementation Details

## C.1    Dataset Descriptions

We conduct experiments on 8 real-world datasets to evaluate the effectiveness of the proposed selective learning, including:

- **ETT** (Electricity Transformer Temperature) [81] contains 7 features of electricity transformer data collected from two separate counties from July 2016 to July 2018. It contains four datasets: **ETTh1, ETTh2, ETTm1, ETTm2**, where ETTh1 and ETTh2 are recorded every hour, and ETTm1 and ETTm2 are recorded every 15 minutes.

- **Electricity** [64] records the hourly electricity consumption data of 321 clients from 2012 to 2014. Each variable represents a client's electricity consumption.

- **Exchange** [64] collects the panel data of daily exchange rates from 1990 to 2016 from 8 countries, including Australia, Britain, Canada, Switzerland, China, Japan, New Zealand, and Singapore.

- **Weather**[64] includes 21 meteorological factors collected every 10 minutes from the weather station of the Max Planck Biogeochemistry Institute in 2020.

- **ILI** (Influenza-Like Illness) [64] includes the weekly recorded patient data from the Centers for Disease Control and Prevention of the United States between 2002 and 2021.

We follow the same data processing and train-validation-test set split protocol used in TimesNet[63], where the train, validation, and test datasets are strictly divided according to chronological order to ensure no data leakage issues. The statistics of the datasets are provided in Table 5.

Table 5: Statistics of the datasets.

| Dataset | Dim | Prediction Length | Dataset Size | Split | Frequency | Domain |
|---------|-----|-------------------|--------------|-------|-----------|--------|
| ETTh1, ETTh2 | 7 | {96, 192, 336, 720} | 14,400 | 6:2:2 | Hourly | Electricity |
| ETTm1, ETTm2 | 7 | {96, 192, 336, 720} | 57,600 | 6:2:2 | 15min | Electricity |
| Exchange | 8 | {96, 192, 336, 720} | 7,588 | 7:1:2 | Daily | Economy |
| Weather | 21 | {96, 192, 336, 720} | 52,696 | 7:1:2 | 10min | Weather |
| Electricity | 321 | {96, 192, 336, 720} | 26,304 | 7:1:2 | Hourly | Electricity |
| ILI | 7 | {24, 36, 48, 60} | 966 | 7:1:2 | Weekly | Health |

## C.2 Baselines

- **Informer** [81] is a Transformer for time series forecasting (TSF) with a ProbSparse self-attention mechanism.
- **Crossformer** [79] utilizes attention to capture both temporal and multivariate correlations.
- **PatchTST** [35] splits the input time series into patches, which serve as input tokens of the Transformer. It proposes a channel-independent strategy.
- **TimesNet** [63] transforms time series into 2D tensors and employs CNN to capture inter- and intra-period dependencies.
- **iTransformer** [29] embeds each series independently to the variate token and applies self-attention to capture multivariate correlations.
- **TimeMixer** [55] is an MLP-based model that captures multi-scale patterns by decomposing time series into different scales and mixing them through MLP layers.

Notably, PatchTST, iTransformer, and TimeMixer are equipped with RevIN [19] to handle the distribution shift issue. Selective learning can provide additional performance gains while maintaining full compatibility with existing normalization techniques.

---

**Algorithm 1** The workflow of selective learning.

---

1: **INPUT:** The model $f(\cdot, \boldsymbol{\theta})$, the training set $\mathcal{D}_{train} = \{(\mathbf{X}_{t-L:t}, \mathbf{X}_{t:t+F})\}_{t=L}^{T-F}$, the estimation model $g(\cdot, \boldsymbol{\phi})$ trained on $\mathcal{D}_{train}$, and the number of iterations $N_{it}$.
2: **OUTPUT:** Optimized model $f(\cdot, \boldsymbol{\theta}_\tau)$.
3: Initialize $f(\cdot, \boldsymbol{\theta}_0)$ and a historical residual archive $S$
4: **for** $\tau$ *in* $\{0, 1, \cdot, N_{it} - 1\}$ **do**
5:    $\hat{\mathbf{X}}_{t:t+F} = f(\mathbf{X}_{t-L:t}; \boldsymbol{\theta}_\tau)$   // Forward
6:    // Calculate the residual and update $S$
7:    $\boldsymbol{\epsilon}_{t:t+F} = \mathbf{X}_{t:t+F} - \hat{\mathbf{X}}_{t:t+F}$
8:    $S \leftarrow \boldsymbol{\epsilon}_{t:t+F}$
9:    // Uncertainty mask
10:    **if** $\tau \geq K$ **then**
11:      // Update the residual entropy once per epoch
12:      **if** $\tau \% K = 0$ **then**
13:        **for** $t$ *in* $\{0, \cdots, T\}$ **do**
14:          Calculate $\hat{H}(\epsilon_t)$ by Eq.(8)
15:        **end for**
16:        $\gamma_u = \text{Top-}r_u\% \ H(\epsilon_t)$   *for* $t \in \{0, \cdots, T-1\}$
17:      **end if**
18:      Calculate $\mathcal{M}_u^{(\tau)}$ by Eq.(11)
19:    **end if**
20:    // Anomaly Mask
21:    Calculate $\hat{S}(X_t)$ by Eq.(13)
22:    $\gamma_a = \text{Top-}r_a\% \ S(X_t)$   *for* $t \in \{0, \cdots, F-1\}$
23:    Calculate $\mathcal{M}_a^{(\tau)}$ by Eq.(14)
24:    $\mathcal{M}^{(\tau)} = \mathcal{M}_u^{(\tau)} \vee \mathcal{M}_a^{(\tau)}$
25:    $\mathcal{L}_{SL} = \frac{1}{N \cdot |\mathcal{M}^{(\tau)}|} \sum_{i=0}^{F-1} ||\mathcal{M}^{(\tau)}(X_{t+i} - f(\mathbf{X}_{t-L:t}; \boldsymbol{\theta}_\tau)_i)||^2$
26:    $\boldsymbol{\theta}_{\tau+1} = \boldsymbol{\theta}_\tau - \eta \nabla_{\boldsymbol{\theta}} \mathcal{L}_{SL}$ // Optimization through selective learning loss
27: **end for**

---

## C.3 Implementation Details

In this section, we provide the implementation details of selective learning. The overall workflow of selective learning is provided in Algorithm 1.

**Uncertainty Mask** We employ a global threshold $\gamma_u$ computed across the entire training sequence $\mathbf{X}_{0:T}$ to determine uncertainty masks. This choice prevents performance degradation caused by

masking normal (but relatively uncertain within specific samples) timesteps when masking ratios increase. For computational efficiency, the threshold is updated only once per epoch. This design additionally mitigates the cold-start issue of uncertainty masking in the first epoch, where inadequate residual entropy estimates would otherwise lead to suboptimal masking decisions.

When processing extremely large datasets that exceed memory capacity for saving full-sequence residuals, we recommend two alternative approaches: (1) adopting per-sample thresholds with conservatively low masking ratios, or (2) implementing a max-heap algorithm for threshold computation. Both solutions maintain computational feasibility while preserving the benefits of uncertain masking.

**Anomaly Mask**    We employ a per-sample threshold $\gamma_a$ computed across the prediction sequence $\mathbf{X}_{t:t+F}$ to determine uncertainty masks. Global anomaly masking over $\mathbf{X}_{0:T}$ is adversely affected by training dynamics (where later samples consistently produce smaller residuals), resulting in suboptimal mask selection.

In this paper, the estimation model $g$ is trained on the entire training set of $f$, diverging from some existing works in other fields that employ held-out sets to train auxiliary models to prevent overfitting in the training set [34]. We prevent overfitting instead by employing a simple model (e.g., DLinear) as $g$, thereby safeguarding the main model's performance. Additionally, unlike other data modalities, the patterns in time-series data often change over time. Consequently, using a holdout set may lead to underfitting of certain patterns in $g$, particularly for datasets with strong non-stationarity. When using a simple model as $g$, it should be trained until full convergence on the training set to avoid underfitting. Additionaly, since the sample partitioning in TSF depends on the sliding window size, each forecasting window size necessitates training a distinct model.

### C.4   Selection of Masking Ratio

The masking ratios are crucial hyperparameters in selective learning. We provide the default masking ratio in Table 6. However, the optimal masking ratio may vary across different models. Users can perform a hyperparameter search around the default masking rate to find the optimal masking ratio. For a new dataset, we recommend a three-stage optimization protocol: (1) optimize the noise masking ratio to achieve peak performance; (2) fix the noise masking ratio and tune the anomaly masking ratio to its optimal value; (3) conduct a local hyperparameter search within the neighborhood of these determined masking ratios.

Additionally, specific masking ratios may also be attempted as initial values:

- $r_u = 10\%, r_a = 10\%$ for stable and high-quality datasets.
- $r_u = 30\%, r_a = 30\%$ for datasets containing certain levels of noise and anomalies.
- $r_a = 90\%$ for highly-volatile dataset.

Table 6: The default masking ratio for the datasets.

| Dataset | ETTh1 | ETTh2 | ETTm1 | ETTm2 | Electricity | Exchange | Weather | ILI |
|---|---|---|---|---|---|---|---|---|
| Uncertainty Mask | 30% | 10% | 20% | 20% | 10% | / | 10% | 10% |
| Anomaly Mask | 30% | 60% | 20% | 50% | 10% | 90% | 20% | 10% |

## D   Running Cost

In this section, we analyze the computational cost introduced by selective learning from the perspective of running time and memory usage.

**Complexity Analysis**    We first theoretically analyze of the time and space complexity of selective learning algorithm.

- **Time Complexity**: Let $B$ be the batch size. For uncertainty mask, residual entropy updates cost $\mathcal{O}(BFN)$ per epoch. The complexity of the anomaly mask depends on the architecture of the estimation model. Taking a linear model as an example, the forward pass of the

estimation model requires $\mathcal{O}(BLN)$ complexity, and the masking process adds $\mathcal{O}(BFN)$. Therefore, the complexity of the anomaly mask is $\mathcal{O}(B(L+F)N)$.

- **Space Complexity**: Storing residuals for $T$ timesteps requires $\mathcal{O}(TFN)$ space.

**Running Time**   We measure the running time per epoch of different models trained without and with selective learning on the ETTh1 and ETTm2 datasets. The results in Table 7 demonstrate that selective learning maintains computational efficiency, adding acceptable training time while achieving significant performance gains.

Table 7: Running cost of selective learning (SL). The results are averaged over 3 runs.

| Method | | TimesNet | | | | iTransformer | | | | TimeMixer | | |
|---|---|---|---|---|---|---|---|---|---|---|---|---|
| Metric | w/o (s/Epoch) | SL (s/Epoch) | Time Inc. (s/Epoch) | MSE Dec. % | w/o (s/Epoch) | SL (s/Epoch) | Time Inc. (s/Epoch) | MSE Dec. % | w/o (s/Epoch) | SL (s/Epoch) | Time Inc. (s/Epoch) | MSE Dec. % |
| ETTh1 96 | 7.2 | 9.0 | 1.8 | 10.6% | 1.8 | 2.0 | 0.2 | 7.7% | 5.0 | 5.4 | 0.4 | 6.9% |
| 192 | 10.2 | 11.0 | 0.8 | 13.0% | 1.9 | 2.2 | 0.3 | 7.0% | 5.1 | 5.6 | 0.5 | 6.8% |
| 336 | 10.2 | 12.4 | 2.2 | 12.9% | 2.1 | 2.3 | 0.2 | 8.1% | 5.1 | 5.8 | 0.7 | 4.0% |
| 720 | 14.1 | 15.7 | 1.6 | 18.9% | 2.2 | 3.1 | 0.9 | 13.6% | 5.3 | 6.5 | 1.2 | 10.5% |
| Avg. | 10.4 | 12.0 | 1.6 | 14.0% | 2.0 | 2.4 | 0.4 | 9.3% | 5.1 | 5.8 | 0.7 | 6.9% |
| ETTm2 96 | 43.2 | 54.5 | 11.3 | 11.5% | 9.7 | 11.0 | 1.3 | 5.1% | 18.1 | 18.8 | 0.7 | 4.7% |
| 192 | 50.1 | 56.6 | 6.5 | 8.1% | 10.2 | 11.2 | 1.0 | 9.0% | 19.2 | 20.0 | 0.8 | 5.6% |
| 336 | 56.3 | 64.2 | 7.9 | 13.1% | 10.9 | 11.8 | 0.9 | 7.2% | 20.5 | 21.1 | 0.6 | 5.7% |
| 720 | 77.4 | 100.5 | 23.1 | 10.0% | 10.3 | 14.2 | 3.9 | 4.2% | 19.1 | 24.0 | 4.9 | 4.3% |
| Avg. | 56.8 | 69.0 | 12.2 | 10.7% | 10.3 | 12.0 | 2.7 | 6.4% | 19.2 | 21.0 | 1.8 | 5.0% |

**Memory Usage**   Our implementation processes historical residuals on the CPU, resulting in merely 2MB of additional GPU memory allocation. Although migrating these operations to the GPU would improve computational throughput, it would increase GPU memory consumption. Additionally, maintaining historical residuals in memory requires approximately $4|\mathcal{D}_{train}|NF$ bytes of RAM (e.g., <0.1GB RAM for ETTh1 and about 7GB for Electricity when $F = 336$).

# E   Discussion

## E.1   Non-generalizable Timesteps vs. Distribution Shift

In recent years, distribution shift in non-stationary time series has been widely studied by the research community [19, 11, 14] and shares conceptual similarities with non-generalizable timestemps. This section compares and contrasts non-generalizable time steps with distribution shift. At their core, both concepts describe a fundamental challenge in deep time series forecasting: the problem of a mismatch between the data a model was trained on and the data it encounters in test, which is typically induced by changes in environmental or exogenous variables

However, the crucial distinction lies in their scale: distribution shift is typically a instance- or segment-level phenomenon, where the statistics of the entire dataset change gradually or abruptly over time. In contrast, a non-generalizable timestep is often a localized, point-level issue. It refers to an individual or a small set of timesteps whose patterns are uncertain or anomalous that they cannot be reliably learned or predicted by the model, even if the overall data distribution remains stable.

Therefore, our proposed selective learning offers a finer-grained solution to prevent models from being affected by non-generalizable data.

## E.2   Dynamic Masking vs. Static Masking

Static masking offers an intuitive implementation approach, for example, training an estimation model to estimate the distribution of each timestep, or employing time series anomaly detection models [40, 71, 66] to target anomalous timesteps. In contrast, dynamic masking adaptively modifies the masked timesteps during training. This approach offers two key advantages:

- Static masking significantly alters the distribution of $\mathcal{D}_{train}$, thereby introducing bias, whereas dynamic masking adapts the mask during training to mitigate this in expectation. Specifically, a given timestep may be masked only during certain training phases and within particular lookback windows, while remaining in other contexts and stages.

- For rare but critical extreme events (e.g., extreme weather)[10, 78] that are less generalizable, dynamic masking first learns the most generalizable timesteps and gradually attempts to learn timesteps previously considered anomalies. Static masking, by contrast, consistently excludes these patterns, resulting in compromised forecasting capacity for extreme events.

We present a comparison between dynamic and static masking in Table 8 using iTransformer, showing that dynamic masking yields consistently better performance, which validates our claims.

Table 8: Comparison of static and dynamic masking.

| Method | Metric | ETTh1 | ETTm2 | Exchange |
|---|---|---|---|---|
| Static masking | MSE | 0.426 | 0.264 | 0.358 |
| | MAE | 0.437 | 0.324 | 0.408 |
| Dynamic masking | MSE | **0.415** | **0.256** | **0.343** |
| | MAE | **0.425** | **0.313** | **0.399** |

### E.3 Capacity of Handling Clean Datasets

If selective learning is employed, then even clean datasets will be masked. This section conducts additional experiments and make discussions to investigate whether selective learning has an impact on model performance on clean datasets.

To evaluate selective learning under clean conditions, we construct a synthetic dataset that is theoretically clean (without any noise or anomalies). An ideal time series without noise or anomalies can be decomposed into trend and periodic components [39, 64]. Accordingly, we synthesize the dataset by combining a linear trend components with daily, weekly, and yearly sinusoidal patterns to generate multivariate time series. For each channel, both the trend slope and the amplitudes of each periodic component are sampled uniformly from specified ranges.

We have conducted experiments on the synthetic dataset using iTransformer as the backbone model. As shown in Table 9, uncertainty masking can reduce model performance on clean datasets (by misidentifying noise), while anomaly masking maintains or even marginally improves performance. This effect stems from our dynamic masking approach for anomalies: When a normal timestep is mistakenly masked in one epoch, it can likely be learned in subsequent epochs.

In summary, we recommend using the anomaly mask on clean datasets, while exercising caution when applying uncertainty masking. Given that real-world time series datasets typically contain a certain level of noise and anomalies, it is therefore beneficial to select an appropriate masking ratio.

Table 9: iTransformer's performance on the synthetic dataset ($L = 336, F = 336$). Better results with selective learning are in **bold**.

| Method | iTransformer | + Uncertainty mask | | | +Anomaly mask | | |
|---|---|---|---|---|---|---|---|
| | | $r_u$=5% | $r_u$=10% | $r_u$=20% | $r_a$=5% | $r_a$=10% | $r_a$=20% |
| MSE | 0.0295 | 0.0475 | 0.1569 | 0.1640 | 0.0299 | **0.0294** | **0.0293** |
| MAE | 0.0838 | **0.0791** | 0.0848 | 0.1004 | **0.0823** | **0.0829** | **0.0833** |

## F Limitations and Future Work

**Beyond Forecasting Task**   Our work currently focuses on time series forecasting tasks. However, our idea can also be applied to other time series analysis tasks, such as imputation and classification [53], guiding the model to focus more on generalizable patterns. While we highlight these potential extensions as promising directions, a thorough investigation of their applicability and effectiveness remains an open question. We leave this exploration for future work.

**Extreme Event Forecasting Capacity**   The dual-masking mechanism may filter out rare extreme events present in the training set. Although dynamic masking can mitigate this effect, the model's

predictive capability for extreme events may still be compromised. For scenarios where extreme events are critically important, we recommend fine-tuning the selective learning-trained model using online learning [59, 36, 22] or test-time adaptation [2, 6, 18, 52] after deployment. For example, SOLID [6] retrieves training samples similar to the current input (including potentially masked ones) to fine-tune the prediction head.

**Pretraining for Time Series Foundation Model**   Recently, time series foundation models (TSFMs) have achieved rapid advancements [62, 9, 30, 1, 46, 41, 16]. Selective learning currently focuses on in-domain forecasting. We leave this as future work. Since the TSFM is trained on samples drawn from a large-scale dataset, we cannot estimate the residual entropy and lower bound at each timestep. As a result, the existing design is not directly compatible with the training of TSFMs. However, thanks to the strong representational capacity of TSFMs, we can train a probabilistic forecasting model (e.g., Chronos [1], MOIRAI [62]) to directly predict the distribution of each timestep, thereby selecting the generalizable timesteps. Future work can further combine the selective learning with some model predictive methods [54] for TSFMs' robustness and efficiency improvement.

# G    Full Experimental Results

## G.1    Full Forecasting Results

The full forecasting results are provided in Table 10 and 11 due to the page limitation of the main text. It can be observed that selective learning significantly improves models' performance in all cases, demonstrating its effectiveness.

## G.2    Full Results of Training Objective Comparison

The full results of the training objective comparison are provided in Table 12 due to the page limitation of the main text. It is evident that selective learning consistently achieves superior performance. These results demonstrate that global alignment over whole sequences, whether in shape or distribution, in the temporal or frequency domain, proves suboptimal, validating the effectiveness of selective learning.

## G.3    Full Ablation Results

The full ablation results are provided in Table 13. The results demonstrate that the model with full selective learning consistently achieves the best performance. Removing either mask leads to significant performance degradation across all four datasets. Additionally, replacing the dual-mask mechanism with random masking reduces model performance to levels comparable to or worse than the unmasked counterparts. This suggests that the effectiveness of selective learning fundamentally stems from our dual-mask mechanism, which directs model attention to *generalizable* timesteps while filtering out *non-generalizable* ones.

# H    Case Study

To showcase the effectiveness of selective learning, we provide supplementary prediction cases of five baselines across five representative datasets in Figure 6. The visualizations clearly show that selective learning can enhance models' forecasting performance and generalizability.

Table 10: Part 1 of the full forecasting results without/with selective learning (SL). Better results are in **bold**. Avg. denotes the average from all prediction lengths, and $\Delta$ denotes the averaged improvements caused by selective learning.

| Method | | Informer | | | | Crossformer | | | | PatchTST | | | |
|---|---|---|---|---|---|---|---|---|---|---|---|---|---|
| | | w/o | | +SL | | w/o | | +SL | | w/o | | +SL | |
| Metric | | MSE | MAE | MSE | MAE | MSE | MAE | MSE | MAE | MSE | MAE | MSE | MAE |
| ETTh1 | 96 | 1.087 | 0.831 | **0.478** | **0.487** | 0.406 | 0.426 | **0.373** | **0.397** | 0.377 | 0.397 | **0.368** | **0.386** |
| | 192 | 1.262 | 0.917 | **0.492** | **0.490** | 0.448 | 0.455 | **0.417** | **0.422** | 0.417 | 0.421 | **0.412** | **0.413** |
| | 336 | 1.379 | 0.952 | **0.503** | **0.526** | 0.460 | 0.466 | **0.448** | **0.443** | 0.448 | 0.442 | **0.433** | **0.426** |
| | 720 | 1.428 | 0.968 | **0.680** | **0.634** | 0.505 | 0.512 | **0.484** | **0.500** | 0.465 | 0.472 | **0.425** | **0.443** |
| | Avg. | 1.289 | 0.917 | **0.538** | **0.534** | 0.455 | 0.465 | **0.431** | **0.441** | 0.427 | 0.433 | **0.410** | **0.417** |
| | $\Delta$ | | | -58.3% | -41.8% | | | -5.33% | -5.22% | | | -4.10% | -3.70% |
| ETTh2 | 96 | 3.107 | 1.475 | **1.516** | **0.937** | 0.717 | 0.579 | **0.474** | **0.463** | 0.311 | 0.367 | **0.296** | **0.353** |
| | 192 | 3.707 | 1.659 | **1.697** | **1.012** | 0.736 | 0.609 | **0.610** | **0.548** | 0.381 | 0.410 | **0.369** | **0.400** |
| | 336 | 2.671 | 1.346 | **1.595** | **0.982** | 0.739 | 0.621 | **0.618** | **0.553** | 0.418 | 0.437 | **0.401** | **0.428** |
| | 720 | 2.543 | 1.348 | **2.065** | **1.175** | 1.113 | 0.784 | **0.935** | **0.707** | 0.443 | 0.464 | **0.431** | **0.452** |
| | Avg. | 3.007 | 1.457 | **1.718** | **1.027** | 0.826 | 0.648 | **0.659** | **0.568** | 0.388 | 0.420 | **0.374** | **0.408** |
| | $\Delta$ | | | -42.9% | -29.5% | | | -20.2% | -12.4% | | | -3.61% | -2.68% |
| ETTm1 | 96 | 0.443 | 0.446 | **0.304** | **0.350** | 0.307 | 0.360 | **0.301** | **0.353** | 0.294 | 0.343 | **0.290** | **0.333** |
| | 192 | 0.618 | 0.573 | **0.344** | **0.374** | 0.366 | 0.404 | **0.344** | **0.381** | 0.339 | 0.373 | **0.333** | **0.359** |
| | 336 | 0.865 | 0.701 | **0.374** | **0.395** | 0.446 | 0.453 | **0.403** | **0.422** | 0.372 | 0.393 | **0.367** | **0.381** |
| | 720 | 0.941 | 0.757 | **0.431** | **0.433** | 0.578 | 0.531 | **0.503** | **0.496** | 0.424 | 0.428 | **0.420** | **0.414** |
| | Avg. | 0.717 | 0.619 | **0.363** | **0.388** | 0.424 | 0.437 | **0.388** | **0.413** | 0.357 | 0.385 | **0.353** | **0.372** |
| | $\Delta$ | | | -49.3% | -37.3% | | | -8.60% | -5.49% | | | -1.33% | -3.32% |
| ETTm2 | 96 | 0.334 | 0.443 | **0.226** | **0.345** | 0.281 | 0.356 | **0.194** | **0.291** | 0.175 | 0.261 | **0.164** | **0.251** |
| | 192 | 0.729 | 0.676 | **0.367** | **0.464** | 0.374 | 0.450 | **0.262** | **0.341** | 0.236 | 0.306 | **0.218** | **0.289** |
| | 336 | 1.416 | 0.955 | **0.595** | **0.610** | 0.676 | 0.582 | **0.391** | **0.430** | 0.293 | 0.347 | **0.266** | **0.322** |
| | 720 | 3.460 | 1.601 | **0.786** | **0.690** | 1.019 | 0.724 | **0.631** | **0.570** | 0.379 | 0.402 | **0.361** | **0.384** |
| | Avg. | 1.485 | 0.919 | **0.494** | **0.527** | 0.588 | 0.528 | **0.370** | **0.408** | 0.271 | 0.329 | **0.252** | **0.312** |
| | $\Delta$ | | | -66.8% | -42.6% | | | -37.1% | -22.7% | | | -6.83% | -5.32% |
| Eletricity | 96 | 0.300 | 0.389 | **0.266** | **0.364** | 0.137 | 0.236 | **0.134** | **0.228** | 0.139 | 0.235 | 0.139 | **0.234** |
| | 192 | 0.311 | 0.399 | **0.286** | **0.384** | 0.162 | 0.260 | **0.147** | **0.241** | 0.153 | 0.248 | **0.152** | **0.245** |
| | 336 | 0.349 | 0.434 | **0.301** | **0.391** | 0.190 | 0.283 | **0.169** | **0.271** | 0.168 | 0.267 | **0.166** | **0.258** |
| | 720 | 0.406 | 0.459 | **0.316** | **0.404** | 0.237 | 0.330 | **0.220** | **0.315** | 0.208 | 0.296 | **0.204** | **0.293** |
| | Avg. | 0.342 | 0.420 | **0.292** | **0.386** | 0.182 | 0.277 | **0.168** | **0.264** | 0.167 | 0.262 | **0.165** | **0.258** |
| | $\Delta$ | | | -14.4% | -8.21% | | | -7.71% | -4.87% | | | -1.05% | -1.53% |
| Exchange | 96 | 0.906 | 0.763 | **0.408** | **0.514** | 0.289 | 0.396 | **0.205** | **0.328** | 0.078 | 0.196 | 0.078 | 0.196 |
| | 192 | 1.291 | 0.908 | **0.621** | **0.638** | 0.527 | 0.558 | **0.355** | **0.447** | 0.161 | 0.290 | **0.157** | **0.287** |
| | 336 | 1.334 | 0.953 | **0.825** | **0.745** | 0.858 | 0.719 | **0.539** | **0.551** | 0.303 | 0.404 | **0.294** | **0.397** |
| | 720 | 2.547 | 1.317 | **1.402** | **0.949** | 1.344 | 0.922 | **1.010** | **0.772** | 0.826 | 0.693 | **0.819** | **0.654** |
| | Avg. | 1.520 | 0.985 | **0.814** | **0.712** | 0.755 | 0.649 | **0.527** | **0.525** | 0.342 | 0.396 | **0.337** | **0.384** |
| | $\Delta$ | | | -46.4% | -27.8% | | | -30.1% | -19.2% | | | -1.46% | -3.10% |
| Weather | 96 | 0.203 | 0.287 | **0.157** | **0.198** | 0.145 | 0.209 | **0.138** | **0.193** | 0.150 | 0.196 | **0.147** | **0.185** |
| | 192 | 0.303 | 0.369 | **0.216** | **0.256** | 0.190 | 0.256 | **0.184** | **0.245** | 0.195 | 0.240 | **0.190** | **0.225** |
| | 336 | 0.351 | 0.373 | **0.270** | **0.305** | 0.252 | 0.306 | **0.231** | **0.285** | 0.246 | 0.280 | **0.243** | **0.267** |
| | 720 | 0.491 | 0.465 | **0.448** | **0.435** | 0.318 | 0.363 | **0.298** | **0.336** | 0.321 | 0.332 | **0.319** | **0.321** |
| | Avg. | 0.337 | 0.374 | **0.273** | **0.299** | 0.226 | 0.284 | **0.213** | **0.265** | 0.228 | 0.262 | **0.225** | **0.250** |
| | $\Delta$ | | | -19.1% | -20.1% | | | -5.97% | -6.61% | | | -1.32% | -4.77% |
| ILI | 24 | 4.689 | 1.466 | **4.587** | **1.434** | 3.595 | 1.265 | **3.006** | **1.150** | 1.900 | 0.868 | **1.755** | **0.856** |
| | 36 | 4.812 | 1.529 | **3.273** | **1.243** | 3.977 | 1.350 | **3.416** | **1.234** | 2.396 | 0.964 | **2.056** | **0.926** |
| | 48 | 4.952 | 1.572 | **3.721** | **1.328** | 3.783 | 1.297 | **3.773** | **1.306** | 1.938 | 0.917 | **1.793** | **0.888** |
| | 60 | 5.358 | 1.608 | **4.405** | **1.434** | 4.571 | 1.457 | **4.527** | **1.450** | 2.070 | 0.933 | **2.014** | **0.911** |
| | Avg. | 4.953 | 1.544 | **3.997** | **1.360** | 3.982 | 1.342 | **3.681** | **1.285** | 2.076 | 0.921 | **1.905** | **0.895** |
| | $\Delta$ | | | -19.3% | -11.9% | | | -7.56% | -4.26% | | | -8.26% | -2.74% |

Table 11: Part 2 of the full forecasting results without/with selective learning (SL). Better results are in **bold**. Avg. denotes the average from all prediction lengths, and $\Delta$ denotes the averaged improvements caused by selective learning.

| Method | | TimesNet | | | | iTransformer | | | | TimeMixer | | | |
| | | w/o | | +SL | | w/o | | +SL | | w/o | | +SL | |
| Metric | | MSE | MAE | MSE | MAE | MSE | MAE | MSE | MAE | MSE | MAE | MSE | MAE |
|---|---|---|---|---|---|---|---|---|---|---|---|---|---|
| ETTh1 | 96 | 0.445 | 0.448 | **0.398** | **0.411** | 0.402 | 0.413 | **0.371** | **0.389** | 0.394 | 0.411 | **0.367** | **0.387** |
| | 192 | 0.476 | 0.472 | **0.414** | **0.426** | 0.445 | 0.440 | **0.414** | **0.420** | 0.440 | 0.442 | **0.410** | **0.415** |
| | 336 | 0.505 | 0.485 | **0.440** | **0.446** | 0.469 | 0.464 | **0.431** | **0.432** | 0.452 | 0.446 | **0.434** | **0.429** |
| | 720 | 0.571 | 0.537 | **0.463** | **0.474** | 0.514 | 0.510 | **0.444** | **0.460** | 0.485 | 0.480 | **0.434** | **0.452** |
| | Avg. | 0.499 | 0.486 | **0.429** | **0.439** | 0.458 | 0.457 | **0.415** | **0.425** | 0.443 | 0.445 | **0.411** | **0.421** |
| | $\Delta$ | | | **-14.0%** | **-9.67%** | | | **-9.29%** | **-6.90%** | | | **-7.11%** | **-5.40%** |
| ETTh2 | 96 | 0.356 | 0.404 | **0.292** | **0.357** | 0.319 | 0.372 | **0.300** | **0.352** | 0.325 | 0.375 | **0.299** | **0.354** |
| | 192 | 0.427 | 0.452 | **0.351** | **0.396** | 0.394 | 0.419 | **0.375** | **0.405** | 0.412 | 0.436 | **0.378** | **0.402** |
| | 336 | 0.450 | 0.467 | **0.389** | **0.423** | 0.429 | 0.445 | **0.408** | **0.433** | 0.430 | 0.451 | **0.405** | **0.428** |
| | 720 | 0.505 | 0.500 | **0.439** | **0.459** | 0.460 | 0.474 | **0.445** | **0.467** | 0.457 | 0.472 | **0.443** | **0.462** |
| | Avg. | 0.435 | 0.456 | **0.368** | **0.409** | 0.401 | 0.428 | **0.382** | **0.414** | 0.406 | 0.434 | **0.381** | **0.412** |
| | $\Delta$ | | | **-15.4%** | **-10.3%** | | | **-4.62%** | **-3.27%** | | | **-6.10%** | **-5.07%** |
| ETTm1 | 96 | 0.329 | 0.375 | **0.298** | **0.342** | 0.305 | 0.358 | **0.295** | **0.342** | 0.298 | 0.350 | **0.287** | **0.337** |
| | 192 | 0.377 | 0.402 | **0.344** | **0.372** | 0.346 | 0.380 | **0.338** | **0.368** | 0.329 | 0.370 | **0.327** | **0.362** |
| | 336 | 0.413 | 0.427 | **0.382** | **0.397** | 0.385 | 0.403 | **0.371** | **0.386** | 0.368 | 0.391 | 0.368 | **0.382** |
| | 720 | 0.464 | 0.453 | **0.419** | **0.424** | 0.446 | 0.441 | **0.423** | **0.422** | 0.431 | 0.426 | **0.422** | **0.416** |
| | Avg. | 0.396 | 0.414 | **0.361** | **0.384** | 0.371 | 0.396 | **0.357** | **0.380** | 0.357 | 0.384 | **0.351** | **0.374** |
| | $\Delta$ | | | **-8.84%** | **-7.36%** | | | **-3.71%** | **-4.05%** | | | **-1.54%** | **-2.60%** |
| ETTm2 | 96 | 0.191 | 0.280 | **0.169** | **0.257** | 0.175 | 0.268 | **0.166** | **0.253** | 0.172 | 0.261 | **0.164** | **0.250** |
| | 192 | 0.246 | 0.314 | **0.226** | **0.298** | 0.244 | 0.315 | **0.220** | **0.291** | 0.233 | 0.305 | **0.220** | **0.289** |
| | 336 | 0.312 | 0.360 | **0.271** | **0.324** | 0.291 | 0.343 | **0.270** | **0.325** | 0.283 | 0.335 | **0.267** | **0.321** |
| | 720 | 0.408 | 0.416 | **0.367** | **0.385** | 0.383 | 0.400 | **0.367** | **0.384** | 0.370 | 0.391 | **0.354** | **0.377** |
| | Avg. | 0.289 | 0.343 | **0.258** | **0.316** | 0.273 | 0.332 | **0.256** | **0.313** | 0.265 | 0.323 | **0.251** | **0.309** |
| | $\Delta$ | | | **-10.7%** | **-7.74%** | | | **-6.40%** | **-5.51%** | | | **-5.01%** | **-4.26%** |
| Eletricity | 96 | 0.184 | 0.289 | **0.177** | **0.281** | 0.134 | 0.227 | **0.132** | **0.223** | 0.135 | 0.229 | **0.133** | **0.228** |
| | 192 | 0.192 | 0.295 | **0.184** | **0.286** | 0.157 | 0.249 | **0.151** | **0.244** | 0.152 | 0.247 | **0.149** | **0.241** |
| | 336 | 0.193 | 0.299 | **0.190** | **0.294** | 0.168 | 0.262 | **0.158** | **0.250** | 0.164 | 0.263 | **0.160** | **0.255** |
| | 720 | 0.222 | 0.320 | **0.214** | **0.313** | 0.197 | 0.290 | **0.187** | **0.279** | 0.201 | 0.297 | **0.196** | **0.290** |
| | Avg. | 0.198 | 0.301 | **0.191** | **0.294** | 0.164 | 0.257 | **0.157** | **0.249** | 0.163 | 0.259 | **0.160** | **0.254** |
| | $\Delta$ | | | **-3.29%** | **-2.33%** | | | **-4.27%** | **-2.92%** | | | **-2.15%** | **-2.12%** |
| Exchange | 96 | 0.109 | 0.239 | **0.091** | **0.218** | 0.088 | 0.211 | **0.082** | **0.203** | 0.080 | 0.199 | **0.078** | **0.197** |
| | 192 | 0.182 | 0.312 | **0.165** | **0.299** | 0.173 | 0.303 | **0.162** | **0.293** | 0.166 | 0.298 | **0.157** | **0.287** |
| | 336 | 0.333 | 0.427 | **0.322** | **0.416** | 0.323 | 0.419 | **0.305** | **0.407** | 0.310 | 0.407 | **0.297** | **0.399** |
| | 720 | 0.904 | 0.736 | **0.875** | **0.720** | 0.870 | 0.720 | **0.821** | **0.693** | 0.834 | 0.694 | **0.808** | **0.693** |
| | Avg. | 0.382 | 0.429 | **0.363** | **0.413** | 0.364 | 0.413 | **0.343** | **0.399** | 0.348 | 0.400 | **0.335** | **0.394** |
| | $\Delta$ | | | **-4.97%** | **-3.73%** | | | **-5.78%** | **-3.45%** | | | **-3.60%** | **-1.38%** |
| Weather | 96 | 0.167 | 0.222 | **0.160** | **0.207** | 0.161 | 0.210 | **0.153** | **0.194** | 0.152 | 0.207 | 0.152 | **0.204** |
| | 192 | 0.218 | 0.266 | **0.212** | **0.255** | 0.206 | 0.249 | **0.197** | **0.237** | 0.197 | 0.253 | **0.194** | **0.243** |
| | 336 | 0.265 | 0.300 | **0.257** | **0.289** | 0.249 | 0.280 | **0.243** | **0.270** | 0.249 | 0.293 | **0.245** | **0.287** |
| | 720 | 0.340 | 0.348 | **0.326** | **0.334** | 0.325 | 0.336 | **0.321** | **0.328** | 0.321 | 0.350 | **0.312** | **0.338** |
| | Avg. | 0.248 | 0.284 | **0.239** | **0.271** | 0.235 | 0.269 | **0.229** | **0.257** | 0.230 | 0.276 | **0.226** | **0.268** |
| | $\Delta$ | | | **-3.54%** | **-4.58%** | | | **-2.87%** | **-4.28%** | | | **-1.74%** | **-2.81%** |
| ILI | 24 | 2.480 | 1.009 | **1.969** | **0.907** | 1.511 | 0.813 | **1.359** | **0.784** | 1.931 | 0.879 | **1.829** | **0.847** |
| | 36 | 2.815 | 1.095 | **2.405** | **0.979** | 1.929 | 0.929 | **1.696** | **0.847** | 2.430 | 0.971 | **2.113** | **0.902** |
| | 48 | 2.436 | 1.008 | **2.351** | **0.941** | 2.054 | 0.931 | **1.857** | **0.892** | 2.135 | 0.931 | **2.051** | **0.903** |
| | 60 | 2.240 | 0.998 | **1.892** | **0.898** | 2.140 | 0.983 | **1.926** | **0.905** | 2.157 | 0.948 | **2.109** | **0.926** |
| | Avg. | 2.493 | 1.028 | **2.154** | **0.931** | 1.909 | 0.914 | **1.710** | **0.857** | 2.163 | 0.932 | **2.026** | **0.895** |
| | $\Delta$ | | | **-13.6%** | **-9.36%** | | | **-10.4%** | **-6.24%** | | | **-6.37%** | **-4.06%** |

Table 12: Full comparison results between selective learning (SL) and other training objectives with iTransformer as backbone. Avg. denotes the averaged results from all prediction lengths. The best results are in **bold**, and the second-best are underlined.

| Loss | | SL | | PS | | FreDF | | TILDE-Q | | MSE | |
|---|---|---|---|---|---|---|---|---|---|---|---|
| Metric | | MSE | MAE | MSE | MAE | MSE | MAE | MSE | MAE | MSE | MAE |
| ETTh1 | 96 | **0.371** | **0.389** | 0.389 | 0.410 | _0.388_ | 0.410 | 0.391 | _0.408_ | 0.402 | 0.413 |
| | 192 | **0.414** | **0.420** | _0.421_ | 0.429 | 0.434 | 0.438 | 0.423 | _0.428_ | 0.445 | 0.440 |
| | 336 | **0.431** | **0.432** | _0.443_ | 0.446 | 0.460 | 0.459 | 0.448 | _0.444_ | 0.469 | 0.464 |
| | 720 | **0.444** | **0.460** | _0.453_ | _0.476_ | 0.516 | 0.513 | 0.467 | 0.477 | 0.514 | 0.510 |
| | Avg. | **0.415** | **0.425** | _0.427_ | 0.440 | 0.450 | 0.455 | 0.432 | _0.439_ | 0.458 | 0.457 |
| ETTm2 | 96 | **0.166** | **0.253** | _0.167_ | _0.255_ | 0.169 | 0.258 | 0.173 | 0.259 | 0.175 | 0.268 |
| | 192 | **0.220** | **0.291** | 0.232 | 0.299 | _0.227_ | _0.298_ | 0.231 | _0.298_ | 0.244 | 0.315 |
| | 336 | **0.270** | **0.325** | 0.287 | 0.336 | _0.274_ | _0.330_ | 0.278 | 0.331 | 0.291 | 0.343 |
| | 720 | **0.367** | **0.384** | _0.370_ | 0.389 | 0.377 | 0.391 | _0.370_ | _0.388_ | 0.383 | 0.400 |
| | Avg. | **0.256** | **0.313** | 0.264 | 0.320 | _0.262_ | _0.319_ | 0.263 | _0.319_ | 0.273 | 0.332 |
| Exchange | 96 | **0.082** | **0.203** | 0.087 | 0.211 | 0.088 | 0.208 | _0.084_ | _0.207_ | 0.088 | 0.211 |
| | 192 | **0.162** | **0.293** | 0.180 | 0.303 | 0.185 | 0.305 | _0.171_ | _0.301_ | 0.173 | 0.303 |
| | 336 | **0.305** | **0.407** | 0.335 | 0.420 | 0.346 | 0.426 | 0.335 | 0.422 | _0.323_ | _0.419_ |
| | 720 | **0.821** | **0.693** | _0.861_ | _0.700_ | 0.886 | 0.712 | 0.884 | 0.724 | 0.870 | 0.720 |
| | Avg. | **0.343** | **0.399** | 0.366 | _0.409_ | 0.376 | 0.413 | 0.369 | 0.414 | _0.364_ | 0.413 |
| Weather | 96 | **0.153** | **0.194** | _0.155_ | 0.199 | 0.159 | 0.207 | _0.155_ | _0.198_ | 0.161 | 0.210 |
| | 192 | **0.197** | **0.237** | 0.200 | 0.241 | 0.204 | 0.249 | _0.199_ | _0.239_ | 0.206 | 0.249 |
| | 336 | **0.243** | **0.270** | 0.250 | 0.281 | 0.260 | 0.292 | _0.249_ | _0.279_ | _0.249_ | 0.280 |
| | 720 | **0.321** | **0.328** | 0.327 | 0.337 | 0.334 | 0.347 | _0.325_ | _0.333_ | _0.325_ | 0.336 |
| | Avg. | **0.229** | **0.257** | 0.233 | 0.265 | 0.239 | 0.274 | _0.232_ | _0.262_ | 0.235 | 0.269 |

Table 13: Full ablation results for selective learning with iTransformer as backbone.

| Design | Prediction Length | ETTh1 | | ETTm2 | | Electricity | | Weather | |
|---|---|---|---|---|---|---|---|---|---|
| | | MSE | MAE | MSE | MAE | MSE | MAE | MSE | MAE |
| **Selective Learning** | 96 | **0.371** | **0.389** | **0.166** | **0.253** | **0.132** | **0.223** | **0.153** | **0.194** |
| | 192 | **0.414** | **0.420** | **0.220** | **0.291** | **0.151** | **0.244** | **0.197** | **0.237** |
| | 336 | **0.431** | **0.432** | **0.270** | **0.325** | **0.158** | **0.250** | **0.243** | **0.270** |
| | 720 | **0.444** | **0.460** | **0.367** | **0.384** | **0.187** | **0.279** | **0.321** | **0.328** |
| | Avg. | **0.415** | **0.425** | **0.257** | **0.315** | **0.157** | **0.249** | **0.229** | **0.257** |
| w/o Uncertainty Mask | 96 | 0.379 | 0.400 | **0.166** | 0.257 | 0.133 | 0.228 | 0.159 | 0.207 |
| | 192 | 0.423 | 0.429 | 0.230 | 0.300 | 0.153 | 0.247 | 0.201 | 0.246 |
| | 336 | 0.448 | 0.447 | 0.287 | 0.335 | 0.165 | 0.261 | 0.246 | 0.280 |
| | 720 | 0.495 | 0.497 | 0.377 | 0.394 | 0.195 | 0.287 | 0.321 | 0.331 |
| | Avg. | 0.436 | 0.443 | 0.265 | 0.322 | 0.162 | 0.256 | 0.232 | 0.266 |
| w/o Anomaly Mask | 96 | 0.388 | 0.403 | 0.177 | 0.261 | 0.135 | 0.228 | 0.158 | 0.206 |
| | 192 | 0.424 | 0.428 | 0.230 | 0.301 | **0.151** | **0.244** | 0.201 | 0.245 |
| | 336 | 0.444 | 0.442 | 0.279 | 0.333 | 0.160 | 0.256 | 0.249 | 0.281 |
| | 720 | 0.468 | 0.477 | 0.376 | 0.396 | 0.189 | **0.279** | 0.326 | 0.334 |
| | Avg. | 0.431 | 0.438 | 0.266 | 0.323 | 0.159 | 0.252 | 0.234 | 0.267 |
| Random Mask | 96 | 0.401 | 0.418 | 0.179 | 0.271 | 0.137 | 0.234 | 0.160 | 0.208 |
| | 192 | 0.452 | 0.452 | 0.240 | 0.312 | 0.158 | 0.256 | 0.203 | 0.249 |
| | 336 | 0.471 | 0.466 | 0.294 | 0.346 | 0.169 | 0.266 | 0.258 | 0.284 |
| | 720 | 0.505 | 0.505 | 0.384 | 0.399 | 0.196 | 0.286 | 0.325 | 0.334 |
| | Avg. | 0.457 | 0.460 | 0.274 | 0.332 | 0.165 | 0.261 | 0.237 | 0.269 |

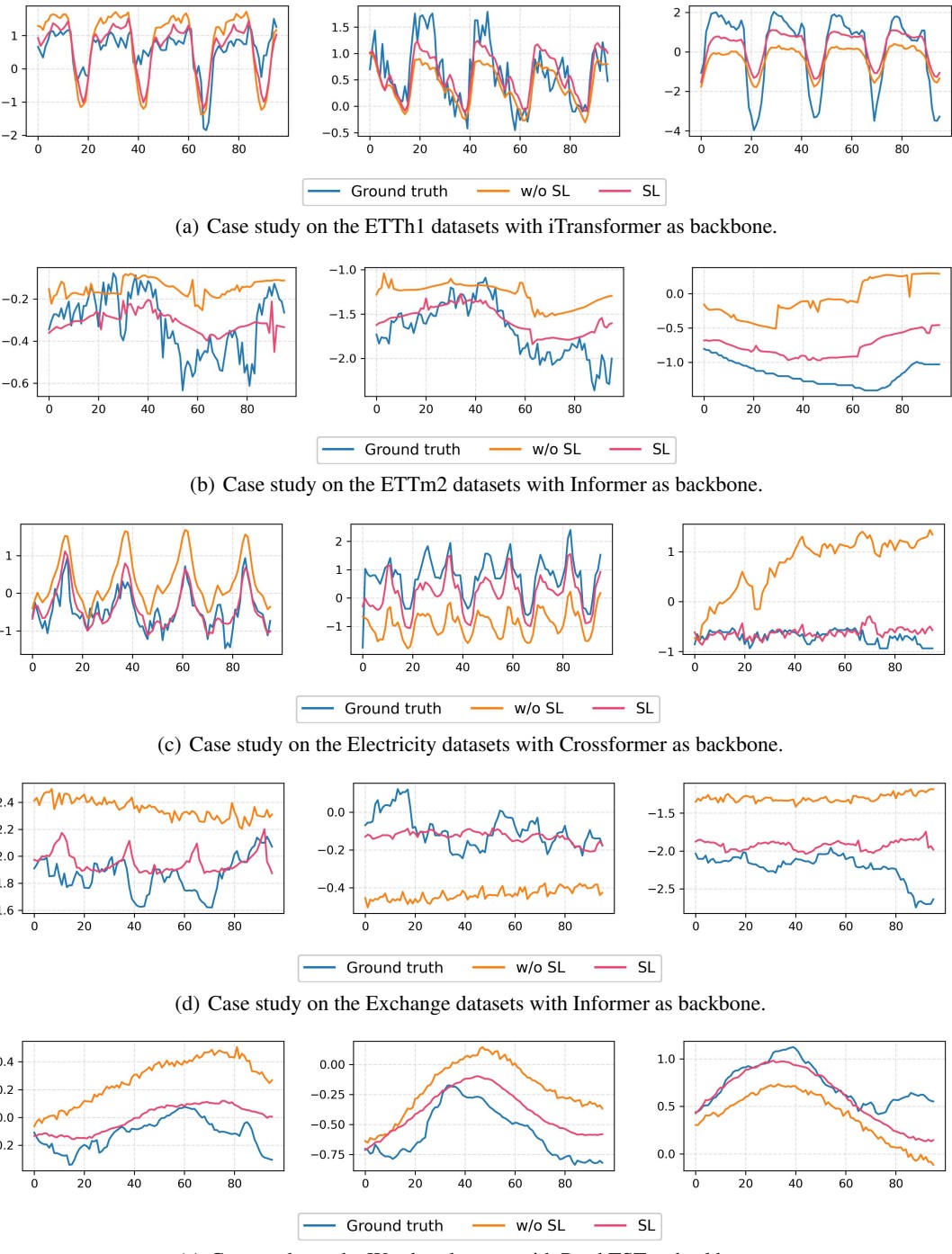

(a) Case study on the ETTh1 datasets with iTransformer as backbone.

(b) Case study on the ETTm2 datasets with Informer as backbone.

(c) Case study on the Electricity datasets with Crossformer as backbone.

(d) Case study on the Exchange datasets with Informer as backbone.

(e) Case study on the Weather datasets with PatchTST as backbone.

Figure 6: Case study results across five datasets.

