# OpenReview forum: "Selective Learning for Deep Time Series Forecasting"
_NeurIPS.cc/2025/Conference — NeurIPS 2025 poster_

### Official Review · Reviewer_vVVG · 2025-06-16

**Clarity:** 2
**Significance:** 3
**Originality:** 3
**Rating:** 4
**Confidence:** 4

**Summary:**

Deep learning models for time series forecasting suffer from overfitting because all future steps, including noisy and abnormal ones, are taken into consideration by the loss function. This work proposes a dual-mask mechanism to filter out noise and anomalies. Noise is filtered by an uncertainty mask derived from residual entropy. Anomalies are filtered by an anomaly mask derived by comparing the residual with that of a pretrained estimation model. Experiments show that the proposed approach can improve the predictive performance of deep models.

**Questions:**

- If assuming that the dataset contains a significant amount of noise and anomalies, does the input side also need adjustment, rather than just at the output end?
- The current approach always filters some timestamps even when the dataset is clean. Is there some way to handle this?
- See my questions in the #Weaknesses Section.

**Ethical Concerns:**

["NO or VERY MINOR ethics concerns only"]

**Final Justification:**

Overall, I recommend accepting this paper and will maintain my initial score of 4. My concerns about computational complexity and input adjustment are addressed. However, there are a few issues that should be addressed or at least discussed in the final version:​​

​The incorporation of entropy:​​ Equations 8–9 currently measure the amplitude of residuals rather than their uncertainty. According to the formal definition of entropy, if all residuals are significant but concentrated, the entropy should be small—contrary to the current measurement. This issue is caused by the assumption of a zero-centered distribution, which is unreasonable, especially when the model is not converged.
​Hyperparameter Sensitivity:​​ The model’s performance is highly sensitive to specific hyperparameters, particularly the uncertainty and anomaly masking ratios.
Clean Data Integration:​​ The current approach does not account for scenarios with clean data, which is another manifestation of hyperparameter sensitivity.

**Limitations:**

Yes

**Paper Formatting Concerns:**

No major formatting issues.

**Quality:**

3

**Strengths And Weaknesses:**

### Strengths

- The idea of using selective learning for TS forecasting is interesting and novel.
- Extensive experiments were conducted to validate the effectiveness of the proposed approach.
- The proposed approach is agnoistic to backbone and can be used as a plug-and-play module.

### Weaknesses
- Some derivations are unclear, for example:
  1. For the uncertainty mask, the inclusion of entropy is confusing. It's hard for me to understand the role of Equations 6 and 7. Moreover, $\epsilon^{(i)} = 100, \forall i$, this timestamp has an entropy of 0 accroading to the defination.
  2. For anomaly mask, also using theoretical lower bound is reasonable, the rational of using another estimation model to detect anomalies needs more explanations, considering lightweight weak models are used here.

- The proposed approach seems to be sensitive to the hyper-parameter masking ratio, which may limit its application:
  1. Best performance is achieved only masking ratios of uncertainty and anomaly masks are jointly tuned, tuning one of them can not get the optimal performance (Figure 3).
  2. Different datasets require different ratios (Table 6).

- Other than runing time estimation in Table 7, detailed theoretical computational efficiency analysis is required.

---

> ### Author Rebuttal · Authors · 2025-07-28
>
> *We sincerely thank Reviewer vVVG for their insightful comments. The following addresses their concerns and provides answers to their questions.*
>
> ---
>
> 1. **Clarifications of the dual-mask mechanism**
>
>    **(a) Uncertainty mask**
>
>    In the uncertainty mask, we use entropy as the measurement of uncertainty. The entropy $H(\epsilon_t)$ quantifies the *predictive uncertainty* of timestep $t$. High entropy indicates unstable predictions across different sliding windows (i.e., the model’s output varies significantly for the same $t$ under different historical contexts).
>
>    **Eq. 6-7 Purpose**: Since calculating entropy using Eq.5 requires the distribution of residuals, we estimate this distribution through Eq.6 with given samples (the likelihood model $l(\boldsymbol{\psi}|\epsilon_t)=p(\epsilon_t|\boldsymbol{\psi})$). And Eq. 7 computes the entropy based on $l(\boldsymbol{\psi}|\epsilon_t)$. We assume $l(\sigma_t|\epsilon_t)=\mathcal N(0,\sigma_t^2)$ for tractability, but other likelihood models could be substituted.
>
>    Our use of Eq. 8 to estimate entropy is based on the assumption that the residuals follow a Gaussian distribution. If all sampled residuals are the same, the estimated entropy would indeed be zero. However, this scenario is highly unlikely with a large sample size. Additionally, we only use the relative magnitude of residual entropy to guide the masking, while the absolute value of the entropy estimation carries no significance.
>
>    **(b) Anomaly mask**
>
>    We adopt another lightweight model as the estimator for two reasons: (1) the theoretical residual lower bound is unattainable, and (2) using the model itself as the estimator yields suboptimal performance (as shown in Figure 4). This is because complex models may have already overfitted during training, making their residual lower bounds ineffective for guiding the masking process. In contrast, using a lightweight model (e.g., DLinear) helps prevent overfitting in the estimation model.
>
> 2. **Masking ratio tuning**
>
>    Thank you for your valuable comment. We agree that joint tuning of uncertainty and anomaly masking ratios is necessary for optimal performance (Fig. 3).
>
>    However, we recommend a three-stage search protocol in Appendix B.4 which reduces the search space and is computationally feasible: (1) optimize the uncertainty masking ratio to achieve peak performance; (2) fix the uncertainty masking ratio and tune the anomaly masking ratio to its optimal value; (3) conduct a local hyperparameter search within the neighborhood of these determined masking ratios.
>
>    Additionally, specific masking ratios may also be attempted as initial values:
>
>    * $r_u=10$%, $r_a=10$% for stable and high-quality datasets.
>    * $r_a=30$%, $r_u=30$% for datasets containing certain levels of noise and anomalies.
>    * $r_a=90$% for highly-volatile dataset.
>
> 3. **Theoretical computational efficiency analysis**
>
>    Thank you for your kind suggestion. Beyond empirical runtime, we provide a theoretical analysis:
>
>    * *Time Complexity*: Let $B$ be the batch size.
>      * Uncertainty Mask: Residual entropy updates cost $\mathcal O(BFN)$ per epoch.
>      * Anomaly Mask: The complexity of the anomaly mask depends on the architecture of the estimation model. Taking a linear model as an example, the forward pass of the estimation model requires $\mathcal O(BLN)$ complexity, and the masking process adds $\mathcal O(BFN)$. Therefore, the complexity of the anomaly mask is $\mathcal O(B(L+F)N)$.
>    * *Memory*: Storing residuals for $T$ timesteps requires $\mathcal O(TFN)$ space.
>
>    Overall, there exists a trade-off between the overhead and performance gain. The overhead (1.6–12.2s/epoch) is justified by significant MSE reductions (up to 66.8%). We will include this analysis in the revised version.
>
> 4. **Does the input side also need adjustment?**
>
>    We appreciate the reviewer's thoughtful comment. Selective learning masks only the output during optimization to steer the model's focus toward generalizable patterns, while keeping the input end unmasked. This design stems from two key considerations: (1) First, since noise and anomalies are inevitable in real-world time series during inference, masking the input during training would undermine model robustness. In contrast, some prior works employ data augmentation (e.g., noise injection) [1] to improve robustness. (2) Second, output (can be viewed as labels) masking excludes non-generalizable patterns from loss computation, therefore enhancing predictive performance.
>
>    This demonstrates an asymmetric treatment between input and output processing: for inputs, we preserve (and even increase) noise to enhance model robustness, whereas for outputs, we aim to reduce noise to prevent overfitting in loss computation.
>
> 5. **Handling clean dataset**
>
>    Thank you for raising this important issue. For theoretically clean datasets where all timesteps are generalizable, omitting masking may be a good choice. However, time series data typically contains inherent redundancy [2], such that moderate masking ratios generally do not compromise model performance.
>
>    Moreover, real-world datasets typically contain varying levels of noise and anomalies, which are the primary cause of model overfitting and the underlying rationale for the effectiveness of selective learning. For high-quality datasets with stable patterns, selecting a relatively small masking ratio can achieve better prediction performance.
>
> ---
>
> **Reference:**
>
> [1] Iglesias, Guillermo, et al. "Data augmentation techniques in time series domain: a survey and taxonomy." *Neural Computing and Applications* 35.14 (2023): 10123-10145.
>
> [2] Lin, Shengsheng, et al. "SparseTSF: modeling long-term time series forecasting with 1k parameters." *Proceedings of the 41st International Conference on Machine Learning*. 2024.
>
> ---
>
> *Thank you for your time and efforts. We hope this has addressed your concerns and answered your questions. Please don’t hesitate to reach out if you have any further questions.*

---

> > ### Comment · Reviewer_vVVG · 2025-08-03
> >
> > Overall, I recommend accepting this paper and will maintain my initial score of 4. However, there are a few issues that should be addressed or at least discussed in the final version:​​
> > 1. ​The incorporation of entropy:​​ Equations 8–9 currently measure the amplitude of residuals rather than their uncertainty. According to the formal definition of entropy, if all residuals are significant but concentrated, the entropy should be small—contrary to the current measurement. This issue is caused by the assumption of a zero-centered distribution, which is unreasonable, especially when the model is not converged.
> > 2. ​Hyperparameter Sensitivity:​​ The model’s performance is highly sensitive to specific hyperparameters, particularly the uncertainty and anomaly masking ratios.
> > 3. Clean Data Integration:​​ The current approach does not account for scenarios with clean data, which is another manifestation of hyperparameter sensitivity.

---

> ### Author Response · Authors · 2025-08-06
> **Response**
>
> We sincerely appreciate your constructive feedbacks. We hope that the additional explanations below will address your concerns.
>
> 1. **Entropy incorporation**
>
>    Thank you for raising this important point. Your observations about the limitations of zero-centered distributions are deeply insightful. We have re-examined the code and found that we computed the variance using the method similar to `residual.var()` (in `basicts/runners/runner_zoo/selective_tsf_runner.py`. See supplementary material), instead of the formulation of Eq.(9). Therefore, according to our implementation, we should calculate $\hat \sigma^2_t$ by:
>    $$
>    \hat \sigma^2_t=\frac{1}{n_t}\sum_{i=1}^{n_t}(\epsilon_t^{(i)}-\bar \epsilon_t)^2,
>    $$
>    which is accordingly under the assumption of $\epsilon_t \sim \mathcal N(\mu_t,\sigma^2_t)$.
>
>    We sincerely apologize for the error and appreciate your valuable contribution in identifying it, which has significantly improved our work. We will correct this in the final version.
>
> 2. **Hyperparameter sensitivity**
>
>    We agree with the reviewer that masking ratios significantly impact model performance, and thank you for pointing this out. To assist practitioners, we included (1) default masking ratios for all benchmark datasets (Appendix B.4), (2) a recommended tuning protocol to efficiently search for optimal values, and (3) empirical effective masking ratio settings that can serve as initialization points for tuning (in our previous response).
>
>    We promise to further *emphasize this guidance and discuss the hyperparameter sensitivity* in the final version.
>
> 3. **Handling clean datasets**
>
>    We appreciate this insightful point and have conducted **additional experiments on a synthetic dataset that is theoretically clean** (without any noise or anomalies) to examine the performance of selective learning on clean datasets.
>
>    An ideal time series without noise or anomalies can be decomposed into trend and periodic components [1,2]. Therefore, we generate the synthetic dataset by combining trend components with daily, weekly, and yearly periodic patterns to form multivariate time series. For each channel, both the trend slope and the amplitudes of each periodic component are sampled uniformly from specified ranges. The core code is provided at the end of the response.
>
>    We have conducted experiments on the synthetic dataset with iTransformer as the backbone model ($L=336, F=336$). As shown in the following table, *uncertainty masking* can reduce model performance on clean datasets (by misidentifying noise), while *anomaly masking* maintains or even marginally improves performance. This effect stems from our dynamic masking approach for anomalies (Line 576 in Appendix D.1): When a normal timestep is mistakenly masked in one epoch, it can likely be learned in subsequent epochs.
>
>    **In summary, we recommend applying the anomaly mask on clean datasets, while using the uncertainty mask with caution.** We will include these results in the final version and open-source the dataset. However, we also acknowledge that providing a theoretical guarantee for no-degradation under clean conditions remains an open question. We will also note this limitation in the discussion.
>
>    ||MSE|MAE|
>    |-|:-:|:-:|
>    |iTransformer|0.0295|0.0838|
>    |+SL (5% Uncertainty mask)  |0.0475| **0.0791** |
>    |+SL (10% Uncertainty mask)|0.1569|0.0848|
>    |+SL (20% Uncertainty mask) |0.1640|0.1004|
>    | +SL (5% Anomaly mask) |0.0299   | **0.0823**|
>    | +SL (10% Anomaly mask) |**0.0294** | **0.0829** |
>    | +SL (20% Anomaly mask) |**0.0293** | **0.0833** |
>
>    **Core code to generate the dataset**
>
>    ```python
>    # generate data for each channel
>        for ch in range(n_channels):
>            daily_amp = np.random.uniform(*daily_amplitude_range)
>            weekly_amp = np.random.uniform(*weekly_amplitude_range)
>            yearly_amp = np.random.uniform(*yearly_amplitude_range)
>            trend_slope = np.random.uniform(*trend_slope_range)
>
>            # generate trend component
>            trend = np.linspace(0, trend_slope * n_points, n_points)
>
>            # generate periodic pattern
>            daily_component = daily_amp * np.sin(2 * np.pi * time_of_day)
>            weekly_component = weekly_amp * np.sin(2 * np.pi * day_of_week)
>            yearly_component = yearly_amp * np.sin(2 * np.pi * day_of_year)
>
>            data[:, ch] = trend + daily_component + weekly_component + yearly_component
>    ```
>
>    **References**
>
>    [1] Cleveland Robert, et al. STL: A seasonal-trend decomposition procedure based on loess. J. Off. Stat, 1990.
>
>    [2] Wu H, et al. Autoformer: Decomposition transformers with auto-correlation for long-term series forecasting. NeurIPS, 2021.
>
> ---
>
> Thank you again for your time and efforts. We hope this has addressed your concerns and we promise to incorporate these analyses and discussions into the final version. Please let us know if you have any further questions.

---

> > ### Comment · Reviewer_vVVG · 2025-08-08
> >
> > Thank you for your response. I have no further questions and hope the final version will incorporate the corresponding changes.

---

### Official Review · Reviewer_93xJ · 2025-06-26

**Clarity:** 3
**Significance:** 3
**Originality:** 3
**Rating:** 5
**Confidence:** 4

**Summary:**

The work proposes a novel selective learning strategy that dynamically ignores spurious timesteps by applying two masks. The uncertainty mask removes any step showing unusually high entropy. Additionally to it the anomaly mask filters out observations that fall outside the lower‑bound forecast generated by a linear model $g$. Experiments show the improved performance and large applicability to several modern TSF approaches.

**Questions:**

- Have the authors tried non-gaussian residuals?
- Both masks rely on noisy residuals before the backbone is warm. The authors mention a warm-up phase, can the authors give more details on that?
- Have the authors tried a held-out validation set to train model $g$? I asked because looks a bit problematic to use the same training set for that too.

**Ethical Concerns:**

["NO or VERY MINOR ethics concerns only"]

**Final Justification:**

I thank you the authors the answers to my concerns. I will keep my score.

**Limitations:**

Yes

**Quality:**

3

**Strengths And Weaknesses:**

Strengths:

- Well written and structured work.
- Easy integration approach, can be plugged in any TSF regression model.
- Interesting idea on filtering timesteps based on uncertainty and anomaly detection.

Weaknesses:

- Entropy mask depends on a unimodal residual fit Gaussian. How SL will behave under a mixture or non-parametric fit?
- Early uncertainty estimation looks a limitation of the approach.

---

> ### Author Rebuttal · Authors · 2025-07-30
>
> *We sincerely thank Reviewer 93xJ for their insightful comments. The following addresses their concerns and provides answers to their questions.*
>
> ---
>
> 1. **Non-Gaussian, mixture, or non-parametric fit of residual in the uncertainty mask**
>
>    Thank you for the valuable feedback. We adopt the Gaussian distribution primarily for computational tractability (since entropy becomes proportional to variance in this case, Eq. 8). Other likelihood models and estimation methods are also theoretically viable. However, in practice, we need to estimate the distribution of *each timestep* in the time series. Using complex distributions (e.g., mixture distributions) or estimation methods (non-parametric methods like KDE) can introduce significant computational overhead. In contrast, under the Gaussian likelihood model, we only need to compute the variance for each timestep in parallel. As shown in Table 1, experiments on eight real-world datasets demonstrate that the Gaussian likelihood model delivers stable performance improvements and is suitable for the vast majority of scenarios.
>
> 2. **Early uncertainty estimation and warm-up phase.**
>
>    Thank you for pointing out this important issue. Conducting uncertainty estimation too early may indeed lead to larger estimation errors. Since the training loss of time series forecasting models typically exhibits its most dramatic decline in the first epoch, we skip uncertainty masking during this initial epoch to avoid introducing excessive variability to the residuals due to training, as outlined in Algorithm 1. In practical implementation, the number of warm-up epochs for the backbone model before applying uncertainty masking can be determined based on the model's convergence behavior.
>
> 3. **Have the authors tried a held-out validation set to train the model?**
>
>    Thank you for the insightful comment. Some existing works indeed employ held-out sets to train auxiliary models [1], primarily to prevent overfitting in the model. In this paper, we employ DLinear (sufficiently simple) as the estimation model $g$ to prevent overfitting on the training set (Line 540), thereby avoiding any degradation in the performance of the main model. Additionally, unlike other data modalities, the patterns in time-series data often change over time. Consequently, using a holdout set may lead to underfitting of certain patterns in $g$, particularly for datasets with strong non-stationarity.
>
>    In the table below, we present a comparative analysis of two sets of results, both from iTransformer on the ETTh1 dataset. The key difference lies in the training data for $g$. The comparative results show that Exp.1 achieves better performance, which validate our claim.
>
>    We will include this analysis in the revised version.
>
>    | ID   | g trained on      |    MSE    |    MAE    |
>    | ---- | ----------------- | :-------: | :-------: |
>    | 1    | full training set | **0.415** | **0.425** |
>    | 2    | 20% held-out set  |   0.425   |   0.437   |
>
>    [1] Mindermann, S., et al. "Prioritized Training on Points that are Learnable, Worth Learning, and not yet Learnt". <i>Proceedings of the 39th International Conference on Machine Learning</i>. 2022.
>
> ---
>
> *Thank you for your time and efforts. We hope this has addressed your concerns and answered your questions. Please don’t hesitate to reach out if you have any further questions.*

---

### Official Review · Reviewer_c9bV · 2025-07-02

**Clarity:** 2
**Significance:** 3
**Originality:** 2
**Rating:** 4
**Confidence:** 3

**Summary:**

This paper proposes Selective Learning, a novel training strategy for deep time series forecasting (TSF). Unlike conventional methods that uniformly optimize all timesteps via MSE loss, this method selectively computes loss only over generalizable timesteps while masking out those identified as uncertain or anomalous. The authors propose a dual-mask mechanism consisting of: an uncertainty mask using residual entropy and an anomaly mask based on residual lower-bound deviation. Extensive experiments on 8 real-world datasets and 6 popular TSF models demonstrate significant improvements, particularly for overfitting-prone architectures.

**Questions:**

See the Strengths And Weaknesses section.

**Ethical Concerns:**

["NO or VERY MINOR ethics concerns only"]

**Final Justification:**

The authors have addressed my concerns in their response. Taking into account both the feedbacks from other reviewers and my own evaluation, I have decided to raise my score.

**Limitations:**

yes

**Paper Formatting Concerns:**

no formatting concerns.

**Quality:**

2

**Strengths And Weaknesses:**

Strengths:

1. The paper addresses a core weakness in current TSF training—overfitting on noisy or anomalous data.
2. The experiments are comprehensive, covering diverse model backbones (Transformer, CNN, MLP), datasets with various periodicity and non-stationarity levels.
3. The paper is very well written with clear problem setup, motivating examples and thoughtful appendices with implementation details and limitations.

Weaknesses and Questions:

1. Concept Drift vs. Anomalies: Many works on non-stationary TSF emphasize concept drift. In realistic scenarios, drift and anomalies often co-exist. How does the proposed method distinguish between the two? Please provide precise definitions and clarify whether “non-generalizable” includes drifted patterns.
2. The proposed approach appears closely related to SIN [1], which also filters out unstable regions. Please clarify the technical differences and include empirical comparison to strengthen the novelty claim.
[1] SIN: Selective and Interpretable Normalization for Long-Term Time Series Forecasting, ICML 2024.
3. The selective learning principle overlaps with ideas from curriculum learning, hard example mining, and prioritized sampling. The paper should provide a clearer conceptual and methodological differences from these approaches in §2.2.
4. The residual entropy and estimated lower-bound both depend on the training dynamics (e.g., optimizer, batch size, learning rate). While Theorem 1 attempts to bound the variance estimation error, empirical sensitivity analysis (e.g., different η, batch sizes) is missing.
5. The use of a second model to estimate residual lower bounds (e.g., DLinear) could introduce biases if the estimation model underfits or overfits. Clarify the generalizability of the estimation model and whether it affects downstream learning in challenging regimes (e.g., high-noise datasets).
6. The method relies on masking ratios (ra, ru) as key hyperparameters. These appear to be set as fixed percentages. This could lead to a situation where on a "clean" dataset, the model is forced to discard perfectly valid timesteps, or on a very "noisy" dataset, it fails to discard enough. How robust is the method for the choice of these ratios, especially across datasets with varying noise levels?
7. The anomaly mask depends on a separate, lightweight estimation model (g(·; φ)). The paper shows that simpler models work better here, which makes sense to avoid overfitting in the estimator itself. However, the performance of the main model is now implicitly dependent on the quality and convergence of this secondary model. How does the performance of Selective Learning degrade if the estimation model g is poorly chosen or poorly trained? Is there a risk of a negative feedback loop where a bad estimator creates bad masks, which further hurts the main model?

---

> ### Author Rebuttal · Authors · 2025-07-30
>
> *We sincerely thank Reviewer c9bV for their insightful comments. The following addresses their concerns and provides answers to their questions.*
>
> ---
>
> 1. **Concept Drift vs. Anomalies**
>
>    Thank you for this valuable comment. In this work, anomaly is defined as a timestep (or a range of timesteps) that significantly deviates from the predicted behavior, in line with domain convention [1]. We acknowledge that anomaly and concept drift often coexist; however, there are distinct differences between the two.
>
>    Concept drift and anomaly both originate from exogenous variable changes but differ in their scales: Concept drift typically reflects distributional shifts over periods (segment or instance level), while anomaly reflects deviations at the timestep level.
>
>    Therefore, anomaly does not necessarily indicate concept drift (as point anomaly may be insufficient to alter the overall distribution). On the other hand, drifted patterns typically contain anomalous timesteps (non-generalizable), but not all timesteps are necessarily non-generalizable.
>
>    We will include this comparison in the revised version to enhance clarity.
>
>    [1] Zamanzadeh Darban, Zahra, et al. "Deep learning for time series anomaly detection: A survey." *ACM Computing Surveys* 57.1 (2024): 1-42.
>
> 2. **Comparison with SIN**
>
>    Thank you for the thoughtful feedback. SIN is indeed an insightful study, but our research demonstrates significant differences in terms of:
>
>    * **Motivation.** Selective learning aims to mitigate model overfitting, while SIN seeks to improve heuristic normalization methods (e.g., RevIN) to better address distribution shift.
>    * **Technically**, SIN is a normalization method where the "selective" targets the statistics during normalization, enabling the model to preserve locally invariant yet globally variant statistics. Selective learning, on the other hand, is a training strategy where the "selective" focuses on the timesteps of time series, allowing the model to concentrate only on generalizable timesteps during optimization. The core technique of SIN is PLS-based covariance maximization, whereas selective learning employs a dual-masking mechanism to filter timesteps.
>
>    We provide an empirical comparison between selective learning and SIN in the table below. The backbone model is iTransformer and the prediction length is 336. Since SIN is not open-sourced, we reproduced it based on the details provided in the paper. The results in the table demonstrate that selective learning consistently outperforms SIN, proving the effectiveness of our work.
>
>    |        | Metric |   ETTh1   |   ETTm2   | Exchange  |
>    | ------ | :----: | :-------: | :-------: | :-------: |
>    | SIN    |  MSE   |   0.481   |   0.296   |   0.326   |
>    |        |  MAE   |   0.473   |   0.346   |   0.420   |
>    | **SL** |  MSE   | **0.431** | **0.270** | **0.305** |
>    |        |  MAE   | **0.432** | **0.325** | **0.407** |
>
>    We promise to incorporate a comparative analysis of this method in the revised version.
>
> 3. **Differences with curriculum learning, hard example mining, and prioritized sampling**
>
>    Thank you for the valuable suggestion. The differences between selective learning and these three paradigms are as follows:
>
>    - *Curriculum learning* progressively increases difficulty (e.g., by expanding prediction lengths in time series tasks), as noted in Line 86.  In contrast, selective learning dynamically filters timesteps based on intrinsic data characteristics (uncertainty and anomalies), rather than relying on predetermined difficulty metrics (e.g., prediction length).
>    - *Hard example mining* and *Prioritized sampling* focus on sample-level selection, prioritizing informative or challenging instances. Our method differs in two critical aspects: (1) We operate at the timestep level, enabling finer-grained optimization of the time series forecasting model. (2) Rather than selecting based on importance or information content, we mask timesteps exhibiting non-generalizable patterns. This strategy aligns with the intrinsic properties of time series data, which are inherently susceptible to noise and anomalies.
>
>    We will incorporate a systematic review of these works and discuss the differences in the Related Work section of the revised version.
>
> 4. **Empirical sensitivity analysis based on Theorem 1**
>
>    Thank you for the valuable suggestion. We conducted sensitivity analyses of the learning rate and batch size on the ETTh1 dataset with iTransformer as backbone. The results are shown in the tables below (prediction length is 336).
>
>    * *Learning rate*: An overly large learning rate (5e-3) can lead to poor estimation performance, while an overly small learning rate (1e-4) may cause the model to get stuck in a local optimum. **Overall, the model is robust to the increase in the learning rate.**
>    * *Batch size*: An overly small batch size (16) can lead to poor estimation performance, while an overly large batch size may also impair model performance (over-smoothed gradient). **Overall, the model is robust to the decrease in the batch size.**
>
>    We will include these analyses in the revised version.
>
>    | Learning rate | 1e-4  |   5e-4    | 1e-3  | 5e-3  |
>    | :-----------: | :---: | :-------: | :---: | :---: |
>    |      MSE      | 0.443 | **0.431** | 0.433 | 0.433 |
>    |      MAE      | 0.445 | **0.432** | 0.434 | 0.436 |
>
>    | Batch size |  16   |  32   |    64     |  128  |
>    | :--------: | :---: | :---: | :-------: | :---: |
>    |    MSE     | 0.433 | 0.433 | **0.431** | 0.436 |
>    |    MAE     | 0.437 | 0.434 | **0.432** | 0.439 |
>
> 5. **Robustness of selective learning to the estimation model**
>
>    Thank you for raising this important concern. We notice that Questions 5 and 7 both concern the estimation model, and we address them together here.
>
>    As shown in Fig. 4, we evaluated the impact of different estimation models (including DLinear, MLP, TimeMixer, and iTransformer) on the final forecasting performance. The results demonstrate that **selective learning remains effective across a range of estimator complexities**, with only marginal variation in final model accuracy. This demonstrates that selective learning is robust to the choice of estimation model.
>
>    Additionally, to avoid instability from a poorly trained estimation model, we recommend training $g$ to full convergence before selective learning begins (Appendix B.3) and using a linear model to avoid overfitting. In this setting, experiments across all eight datasets (Tables 1&2) show **no performance degradation** attributable to the estimation model. This demonstrates the effectiveness of the current estimation setting in most scenarios (including high-noisy datasets such as ETT).
>
>   Due to the simplicity and generalizability of the linear estimation model, we believe the likelihood of such negative feedback loops is minimal in practice. Nonetheless, we will add an analysis in the revision discussing the robustness of selective learning.
>
> 6. **Choice of masking ratios**
>
>    We acknowledge that the masking ratios are crucial hyperparameters in selective learning. However, for datasets with different characteristics (e.g., high-quality, noisy), we can search for appropriate masking ratios to achieve optimal results (see Appendix B.4 for details).
>
> To facilitate this process, we provide empirically effective masking ratio settings that can serve as initialization points for tuning:
>
>    * $r_u=10$%, $r_a=10$% for stable and high-quality datasets.
>    * $r_a=30$%, $r_u=30$% for datasets containing certain levels of noise and anomalies.
>    * $r_a=90$% for highly-volatile dataset.
>
>    Noise and anomalies are often unevenly distributed within a dataset. To mitigate the risk of over-discarding clean data or under-discarding noisy data, we adopt a global masking strategy (Line 525), applying masking to all timesteps across samples at an appropriate masking ratio. Additionally, dynamic masking prevents excessive discarding of normal timesteps (Line 576). When a normal timestep is mistakenly discarded in one epoch, it can likely be learned in subsequent epochs.
>
> ---
>
> *Thank you for your time and efforts. We hope this has addressed your concerns and answered your questions. Please don’t hesitate to reach out if you have any further questions.*

---

> > ### Comment · Reviewer_c9bV · 2025-08-05
> >
> > Thank you very much for your response and the effort you put into it. Your detailed explanations and the additional experiments have largely addressed my concerns, so I have decided to raise my score.

---

> > > ### Author Response · Authors · 2025-08-06
> > >
> > > We are glad that our responses addressed your concerns. Your constructive feedback helped us improve the quality of this work. We will incorporate the corresponding revisions into the final manuscript. Thank you again for your comments.

---

### Official Review · Reviewer_3Yh3 · 2025-07-02

**Clarity:** 4
**Significance:** 3
**Originality:** 3
**Rating:** 5
**Confidence:** 4

**Summary:**

This paper introduces selective learning for deep time series forecasting. Selective learning only calculate the MSE loss over a subset of timesteps, guiding the model to focus on generalizable timesteps while disregarding non-generalizable ones. Technically, the paper propose a dual-mask mechanism (including an uncertainty mask and an anomaly mask) to target timesteps. Extensive experiments across eight real-world datasets demonstrate that selective learning can significantly improve the performance for state-of-the-art deep models.

**Questions:**

Q1. Although selective learning demonstrates reduced MSE and MAE metrics, could the masking strategy potentially introduce bias to models?

Q2. In Figure 5, why does it present results of Informer for 30 epochs while other models for only 10 epochs?

Q3.In Assumption 4, what does $\au$ stand for?

Q4. The paper compares selective learning with other training objectives (e.g., PS loss and FreDF). If I understand correctly, selective learning appears theoretically compatible with these objectives. Could the idea of selective learning be combined with existing objectives to further enhance their performance?

**Ethical Concerns:**

["NO or VERY MINOR ethics concerns only"]

**Final Justification:**

My major concerns on the experiments and generalization have been well addressed in the rebuttal. And after reading other reviewers' comments and authors' responses, I decide to keep the score and would like to recommend the acceptance of the paper.

**Limitations:**

Yes

**Quality:**

3

**Strengths And Weaknesses:**

Strengths:

S1. This paper proposes a novel training strategy for deep time series forecasting and offers a valuable insight: optimizing a generalizable subset of the time series is more effective than optimizing the entire sequence due to the inherent vulnerability of time series to anomalies and noise.

S2. Extensive experiments and rigorous analysis demonstrate the effectiveness of selective learning. The performance gain from selective learning is significant, particularly for those models prone to overfitting (e.g., Informer, Crossformer).

S3. The paper is well-written with clear explainations and is easy to follow.

Weaknesses:

W1. In lines 170-176, the authors state that the static masking approach is suboptimal. However, the experiment does not include a comparative analysis between the current dual-masking mechanism and static masking approach.

W2. While the idea of selective learning is general, the current technical implementation of selective learning is confined to in-domain prediction task and cannot be applied to the training of time series foundation model.

---

> ### Author Rebuttal · Authors · 2025-07-28
>
> *We sincerely thank Reviewer kupb for their insightful comments. The following addresses their concerns and provides answers to their questions.*
>
> ---
>
> 1. **Comparative analysis between dynamic masking and static masking.**
>
>    Thank you for your valuable suggestion. We conduct an additional comparative analysis between dynamic masking and static masking with iTransformer as the backbone. As shown in the table below, dynamic masking achieves better performance than static masking, validating our claims. We will include this comparative analysis in the revised version.
>
>    | Method          | Metric |   ETTh1   |   ETTm2   | Exchange  |
>    | :-------------- | :----: | :-------: | :-------: | :-------: |
>    | Static masking  |  MSE   |   0.426   |   0.264   |   0.358   |
>    |                 |  MAE   |   0.437   |   0.324   |   0.408   |
>    | Dynamic masking |  MSE   | **0.415** | **0.256** | **0.343** |
>    |                 |  MAE   | **0.425** | **0.313** | **0.399** |
>
> 2. **The scope of our work**
>
>    The scope of our work focuses on in-domain time series forecasting. Since uncertainty masking requires multiple prediction samples of timesteps within an epoch, the current masking mechanism cannot be directly applied to the pretraining of foundation models. However, the high-level idea of selective learning is not inherently tied to specific masking implementations and can be adapted to broader task scenarios. We will explore masking mechanisms for TSFMs in future work.
>
> 3. **Could the masking strategy potentially introduce bias to models?**
>
>    Thank you for raising this important point. Selective learning improves model generalization through masking (reducing the occurrence of non-generalizable patterns), which inevitably introduces bias (e.g., degraded performance in predicting extreme events). This represents a trade-off. However, techniques like online learning and test-time adaptation can help enhance the prediction performance for extreme events that might be overlooked during training.
>
> 4. **Issues about Figure 5**
>
>    Due to severe overfitting in Crossformer and TimeMixer (Fig. 5), plotting more epochs would make the selective learning curve appear nearly flat compared to the original curves. Therefore, we only display the results of Crossformer and TimeMixer for 10 epochs.
>
> 5. **Explanation of Assumption 4**
>
>    We sincerely apologize for the typographical error that caused confusion. The correct version of Eq.17 in Assumption 4 should be:
>    $$
>    ||\nabla_{\boldsymbol{\theta}_\tau} \mathcal L||<G, \quad \forall \tau\in \mathbb{Z}^{+},
>    $$
>    where $\tau$ is the number of iteration. We will correct this in the revised version.
>
> 6. **Could the idea of selective learning be combined with existing training objectives?**
>
>    Selective learning can indeed be combined with certain training objectives. The core idea of selective learning is to mask non-generalizable timesteps, which grants it strong compatibility. For example, the table (prediction length is 336) below demonstrates that selective learning can further improve the performance of the FreDF loss.
>
>    |              |    MSE    |    MAE    |
>    | ------------ | :-------: | :-------: |
>    | iTransformer |   0.469   |   0.464   |
>    | + FreDF      |   0.460   |   0.459   |
>    | + FreDF + SL | **0.436** | **0.439** |
>
> ---
>
> *Thank you for your time and efforts. We hope this has addressed your concerns and answered your questions. Please don’t hesitate to reach out if you have any further questions.*

---

> > ### Comment · Reviewer_3Yh3 · 2025-08-06
> >
> > My major concerns on the experiments and generalization have been well addressed in the rebuttal. And after reading other reviewers' comments and authors' responses, I decide to keep my positive score.

---

> > > ### Author Response · Authors · 2025-08-06
> > >
> > > Thank you for providing the insightful review, which has significantly contributed to improving our manuscript. We will incorporate the experimental results and corresponding discussions into the revised version.

---

### Note · Authors · 2025-08-12

We sincerely thank all reviewers for their constructive reviews, which greatly help us improve our paper.

The reviewers generally viewed our work positively, noting that it is **"novel and interesting"** (Reviewer 3Yh3, 93xJ, vVVG), **"easy to integrate and plug-and-play"** (Reviewer 93xJ, vVVG), **"offers a valuable insight"** (Reviewer 3Yh3), and **"addresses a core weakness in TSF"** (Reviewer c9bV). They also found **"the paper well-written"** (Reviewer 3Yh3, c9bV, 93xJ), **"experiments extensive and comprehensive"** (Reviewer 3Yh3, c9bV, vVVG), and **"performance gains significant"** (Reviewer 3Yh3).

In the final version, we will incorporate the following revisions based on reviewers’ valuable suggestions:
* **Anomaly vs. concept drift (Reviewer c9bV).** We have clarified the definition of anomaly and its difference from concept drift in their scales (timestep vs. segment level). We will add this explanation in the revision.
* **Clarify differences among curriculum learning, hard example mining, prioritized sampling, and normalization methods for concept drift (Reviewer c9bV).** We have clarified key differences in motivation and technical design in the rebuttal, and we will elaborate in Related Work.
* **Add an analysis on the training data of the model g (Review 93xJ).** We have provided discussions and experimental results on using a held-out set. We will include these results and provide a detailed discussion on how to avoid over/underfitting in *g*.
* **Additional experiments: 1) static masking comparison (Reviewer 3Yh3); 2) sensitivity analysis regarding Theorem 1 (Reviewer c9bV); 3) results on synthetic data to evaluate performance on clean data (Reviewer vVVG).** We have conducted these experiments to address the concerns in the rebuttal and will include the results in the revision.
* **Analyze the robustness of selective learning regarding the choice of the masking ratios and model g (Reviewer c9bV, vVVG).** We will further analyze the robustness, add more guidelines for selecting appropriate masking ratios/model *g*, and update Appendix B.4 with empirically effective masking ratios from our rebuttal.
* **Add theoretical computational efficiency analysis (Reviewer vVVG).** We have analyzed the computational complexity in our rebuttal and will include it in Appendix C.
* **Eq. (9) and (17) will be revised accordingly (Reviewer vVVG, 3Yh3).** We thank the reviewers for noting these issues; all fixes will be made as specified in our response.

---

### Decision · Program_Chairs · 2025-09-17

**Decision:**

Accept (poster)

**Comment:**

This well-written paper proposes a new selective learning approach to training time series forecasting models that smartly focuses on more generalizable parts of the training data. It has been evaluated by 4 knowledgeable reviewers who all agreed in their final evaluations that it is acceptable for NeurIPS (2 straight accept scores, 2 marginal accepts), despite noted weaknesses (most of which the authors have managed to address in the rebuttal and discussion with the reviewers). The authors are strongly encouraged to carefully reflect on the feedback received in the final revision of the paper.